

# The seamless and multi-model coupling between atmosphere, land, hydrology, ocean, waves and sea-ice models based on SURFEX surface model using OASIS3-MCT

Aurore Voldoire[1], Bertrand Decharme[1], Joris Pianezze[2], Cindy Lebeaupin Brossier[1], Florence Sevault[1],

Léo Seyfried[3], Valérie Garnier[2], Soline Bielli[4], Sophie Valcke[5], Antoinette Alias[1], Mickael Accensi[2],

Fabrice Ardhuin[2], Marie-Noëlle Bouin[1,2], Véronique Ducrocq[1], Stéphanie Faroux[1], Hervé Giordani[1],

Fabien Léger[1], Patrick Marsaleix[3], Romain Rainaud[1], Jean-Luc Redelsperger[2], Evelyne Richard[3],

Sébastien Riette[1]

[1]Centre National de Recherches Météorologiques (CNRM, Météo-France/CNRS UMR3589), Toulouse, France

[2]Laboratoire d'Océanographie Physique et Spatiale (LOPS, CNRS UMR6523/Ifremer/IRD/UBO), Brest, France

[3]Laboratoire d'Aérologie (LA, CNRS UMR5560/Université Toulouse), Toulouse, France

[4]Laboratoire de l'Atmosphère et des Cyclones, (LACy, CNRS UMR8105/Université de la Réunion/Météo-France), Saint Denis de la Réunion, France

[5]Centre Européen de Recherche et de Formation Avancée en Calcul Scientifique (CERFACS/CNRS UMR 5318), Toulouse, France

*Correspondence to*: Aurore Voldoire (aurore.voldoire@meteo.fr)

**Abstract.** This study presents the principles of the new coupling interface based on the SURFEX multi-surface model and the OASIS3-MCT coupler. As SURFEX can be plugged into several atmospheric models, it can be used in a wide range of applications, from global and regional coupled climate systems to high-resolution Numerical Weather Prediction systems or very fine scale systems dedicated to process studies. The objective of this development is to build and share a common structure for all these applications for the atmosphere-surface coupling, between atmospheric models and ocean, ice, hydrology, and wave models. The numerical and physical principles of the SURFEX interface between the different component models are described, and the different coupled systems into which the SURFEX OASIS3-MCT-based coupling interface is already applied are presented.



# 1 Introduction

In late 80's, the first coupled systems assembling atmosphere and ocean models were developed for climate-scale studies. The interactions between the atmosphere and the ocean need indeed to be properly represented when considering the climate system on long time-scales. Seasonal forecasting also called for such coupled models. Indeed, El Niño Southern Oscillation (ENSO) is one of the main process that drives the predictability at the seasonal scale and is in essence a coupled
process that can not be simulated either by an atmospheric model nor by an oceanic model alone. In the last decade, General Circulation Models (GCM) have progressively evolved to become Earth System Models (ESM) by including other components such as sea-ice, carbon cycle, chemistry and continental hydrology. In the meantime, Regional Climate System Models (RCSM) coupling atmosphere and ocean limited-area models have been developed. They have shown their usefulness in increasing the reliability of regional climate information in areas where local and complex interactions and
feedbacks between the different components of the system are important such as in the Mediterranean region (MED-CORDEX initiative; Ruti et al, 2016).

Higher-resolution (<~5km-resolution) modelling systems generally used for numerical weather prediction (NWP) and fine scale process studies are rarely coupled systems, arguing that the ocean evolves on much slower timescales than the atmosphere. However, there are some exceptions for which an immediate response of the ocean to the atmospheric weather
is observed, with short-time, intense and localized interactions between the two components, as such encountered during extreme weather events (e.g., tropical cyclones, strong winds and storms at mid-latitudes, heavy precipitation events).

The development of ocean-atmosphere high-resolution coupled systems started at the beginning of the 2000's (e.g., Pullen et al., 2003; Rein et al., 2004; Loglisci et al., 2004). Nowadays, the development of high-resolution, short to medium range coupled prediction systems is still challenging, but several groups have undertaken it, based on coupling methods with
different levels of sophistication and targeting a large range of applications depending on their interests (Brassington et al., 2015; Heinzeller et al., 2016).

From the technical point of view, different approaches can be followed to implement the coupling between existing model components. Fully embedded coupling (i.e. assembling components into one executable in a hard-coded way, using or not classes and methods from coupling toolkits such as the Model Coupling Toolkit, Larson et al., 2005) can be chosen but
restricts the modularity of the coupled system and the reuse of its components. Therefore many coupled systems are built using higher-level coupling technologies specifically developed for that purpose. These coupling technologies can be roughly divided into two categories, both including the ability to communicate data between components, regrid data, and manage the time evolution of the model integration. Coupling using a concurrent multiple executable approach (e.g. OASIS3-MCT, Valcke et al., 2015) requires a minimal amount of modifications in existing component codes but limits the
ways they can be run on available computing resources, which can hinder performance. Coupling via component-level interfaces within one integrated application (e.g. the Earth System Modeling Framework, ESMF,



http://www.earthsystemmodeling.org, Collins et al., 2005, Theurich et al, 2016) requires users to split components into initialise, run, and finalise parts, and limit the places where data exchanges can happen; however, because components can be run sequentially or concurrently, there are additional opportunities for performance optimisation.

An example of coupled system using the ESMF coupler is the *Coupled Ocean-Atmosphere Mesoscale Prediction System* (COAMPS®) developed by the Naval Research Laboratory (NRL) and run in operations by the U.S. Department of Defense. In that case, an ESMF coupler component receives variables from the ocean model upper level and from the lowest atmospheric model level to compute the air-sea exchanges on an intermediate grid, and then sends them back to the two component models. COAMPS® also includes nesting capability in the two components and a coupled data assimilation

scheme. This system was used in particular over the Adriatic Sea and the Ligurian Sea areas to evaluate the air-sea interactions during strong wind events, Bora and Mistral, respectively, with resolutions up to 4km in the atmosphere and 2km in the ocean model (Pullen et al., 2006, 2007; Small et al., 2011, 2012). A configuration of COAMPS, named COAMPS-TC, is specifically designed to improve tropical cyclone forecasts (Doyle et al. 2014).

       The *Coupled Ocean-Atmosphere-Wave-Sediment-Transport* system (COAWST, Warner et al., 2010) uses directly

the lower level *Model Coupling Toolkit* (MCT, Larson et al., 2005) library and the *Spherical Coordinate Remapping Interpolation Package* (SCRIP, Jones, 1999) interpolation method and assembles into one executable the *Weather Research and Forecasting* (WRF, Skamarock et al., 2008) atmospheric model, the *Regional Ocean Modeling System* (ROMS, Shchepetkin and McWilliams, 2005) oceanic model, the *Simulating WAves Nearshore* (SWAN, Booij et al., 1999) wave model and the *Community Sediment Transport Model* (CSTMS, Warner et al., 2008). COAWST was notably used with high

resolutions (up to 3 km for atmosphere and up to 1 km for the ocean and wave models) over several places in the Mediterranean region (Renault et al., 2012; Carniel et al., 2016; Ricchi et al., 2016). These studies highlight that high-resolution coupling improves significantly the simulation results.

       This paper presents the development of a standard coupling interface in the SURFEX surface model (Masson et al., 2013) based on the OASIS3-MCT coupler (Valcke et al., 2015). SURFEX is a fully externalized surface model that

represents the evolution of four surface types (land, town, ocean and inland waters) and of surface-atmosphere fluxes. This surface model can be used in stand-alone mode, i.e. driven by a prescribed atmospheric state, or embedded in an atmospheric model. The introduction of a standard OASIS3-MCT coupling interface allows the surface model to be linked to sophisticated models also for the other components, such as ocean, ice, wave and hydrological models when needed. This is particularly mandatory when the 0D or 1D surface parameterizations implemented in SURFEX are not valid or fail. A

standard coupling interface has been preferred to embedded couplings as it helps the use of various models for each component depending on the application and fosters interoperability between the models. It allows also to benefit more easily and rapidly from modelling advances achieved by each model community. SURFEX is used in the following atmospheric models: ARPEGE (Courtier et al., 1991), the Météo-France global model for NWP and climate (Déqué et al.,



1994); ALADIN (Fisher et al., 2005), the limited area configuration of ARPEGE initially developed for NWP and now used for regional climate (Spirinov et al., 2005; Radu et al., 2008); AROME, the non-hydrostatic limited area model in operation at Météo-France (Seity et al., 2011) and the derived HARMONIE configuration used by the HIRLAM consortium; MESO-NH (Lafore et al., 1998), a research oriented non-hydrostatic atmospheric model developed jointly by the Laboratoire d'Aérologie and CNRM. Developing a standard interface in SURFEX to couple it with ocean, wave and hydrological models thus means that all these atmospheric models can be assembled into various coupled systems.

OASIS3-MCT has been chosen as coupling interface in SURFEX since this coupling library was already used for coupling previous versions of ARPEGE-climate with the ocean model NEMO (Voldoire et al., 2013). SURFEX, with its OASIS3-MCT interface, can be run for different time-scales and can be used in stand-alone mode or embedded in atmospheric models to be coupled to other models, such as ocean, ice, wave and hydrology models. The standard interface development was initiated in SURFEX V7.3. This development has also been back-phased in former version of SURFEX, in order to be used with other atmospheric models using a previous versions of SURFEX. The standard coupling interface with OASIS3-MCT is now part of the SURFEX open-source code suite since release v8, except for the the additional code for exchanges with a wave model which is available in the following version.

This paper presents the numerical and physical principles of the standard coupling interface in section 2. Section 3 provides several examples illustrating use of the interface, ranging from climate applications to process oriented studies with mesoscale models. Conclusions and perspectives are given in section 4.

## 2 Principles of the SURFEX-based coupling using OASIS3-MCT

### 2.1 SURFEX brief description

A complete description of SURFEX can be found in Masson et al. (2013). In summary, SURFEX computes the surface prognostic variables (surface temperature, radiative temperature, roughness length, albedo, emissivity) and fluxes (evaporation and evapotranspiration, sensible and latent heat fluxes, wind stress) taking into account the evolution of four types of surfaces: land, water, ocean and town. To do so, the SURFEX model includes various schemes:

For the land surface type, the "Interactions between Soil, Biosphere, and Atmosphere" (ISBA) scheme (Noilhan and Planton, 1989) is used. Several parameterisations are available in ISBA to represent the evolution of continental natural surfaces including bare soils, rocks, permanent snow, glaciers, natural vegetation and agricultural landscapes;

Fluxes over sea and ocean are obtained with bulk parameterizations, either direct like Louis (1979)'s scheme or iterative like the "Exchange Coefficients from Unified Multicampaign Estimates" (ECUME) (Belamari, 2005, Belamari and Pirani, 2007) or the "Coupled Ocean-Atmosphere Response Experiment" (COARE) (Fairall et al., 2003) parameterizations;

Inland water (including lakes and rivers) is treated with the Charnock (1955)'s formulation or with the FLAKE scheme (Mironov, 2010);



When it is not considered as rocks, urban (town) surface (including buildings, roads and transportation infrastructures, and gardens) is modeled using the Town Energy Budget (TEB) scheme (Masson, 2000).

The surface-atmosphere fluxes for each surface types are then aggregated over the model grid-mesh, according to the fraction of the four types of surface in the grid-mesh. The averaged value (F) over the grid-mesh is thus given by:

$$F = c_{nature} F_{nature} + c_{ocean} F_{ocean} + c_{lake} F_{lake} + c_{town} F_{town} \qquad (1)$$

$F_{nature}$, $F_{ocean}$, $F_{lake}$, $F_{town}$ and $c_{nature}$, $c_{ocean}$, $c_{lake}$, $c_{town}$ are the surface-atmosphere fluxes and fraction of each type within the grid-mesh for nature, ocean, inland water and town, respectively.

### 2.2 Interfacing of the SURFEX surface modelling platform with OASIS3-MCT

The standard coupling interface introduced in SURFEX, relies on the OASIS3-MCT library (Valcke et al., 2015). OASIS is developed by CERFACS since 1991 and is now interfaced with MCT (Larson et al., 2005). OASIS3-MCT is a
coupling library which main function is to exchange and interpolate fields between different codes modeling the different components of the system. Thanks to MCT, all transformations are executed in parallel on the source or target processes and parallel coupling exchanges are executed via Message Passing Interface (MPI) directly between the components (indeed the OASIS3-MCT coupling library does not need dedicated processes as was the case for the previous OASIS3 version, Valcke 2013). Here, the development consisted in including OASIS3-MCT subroutine calls in the SURFEX v8 source code (in the
subroutines indicated in italic between parentheses) as follows :

initialization (sfx_oasis_init) and namelist reading (sfx_oasis_read_nam);

multi-process partition definition and listing of the exchanged fields (sfx_oasis_define);

receiving (sfx_oasis_recv) and sending (sfx_oasis_send) of the coupling fields;

finalization (sfx_oasis_end).

Steps 1, 2 and 4, which correspond to preparation and closure of the coupling, are either called by SURFEX OFFLINE main program when SURFEX runs alone or by the atmospheric model when SURFEX is integrated into one (because the parallelisation information to be sent to OASIS is managed by the atmospheric model). Receiving and sending actions are directly called by SURFEX in both modes.

For all coupling exchanges, the source and the target components, the coupling frequency of the exchanges, and the
interpolation method between the different grids are chosen by the user and specified in an external text file, the "namcouple". Before running the coupled system, the user has to prepare the coupling restart files, if some are needed given the coupling algorithm. The user can also prepare the files containing the weights for interpolation if he does not want to rely on OASIS3-MCT to generate them automatically. The coupling grid definition files can also be either prepared by the user before the run or automatically generated using the SURFEX PREP tool that can also generate SURFEX restarts files.



However, the grids and masks automatically defined may be not suitable for non global coupled model configurations (see section 3.2 and 3.3).

### 2.3 Flow chart of the coupling exchanges

The complete flow chart of the coupling exchanges between SURFEX (embedded or not in an atmospheric model - ATM), an ocean model (OCE) with sea-ice (ICE), a wave model (WAV) and an hydrological model (HYD) is shown on

Figure 1. The coupling with such components is intended to be generic and not dependent on the models used for OCE, ICE, WAV and HYD. This has only been tested for OCE, as both NEMO and SYMPHONIE have been coupled to SURFEX. However, even if few coupling fields may be missing in the case of coupling a new model for ICE, HYD or WAV, the general flow chart remains unchanged. Adding a new coupling field is also relatively straightforward. In this paper, we do not discuss the exchanges between SURFEX and the ATM component, as they are not done through OASIS; they are

described in detail in Masson et al. (2013).

### 2.3.1 OCE-ICE-SURFEX

The atmosphere exchanges heat, water and momentum with the ocean. These exchanges are expressed as the net solar heat flux ($Q_{sol}$), the non-solar heat flux ($Q_{ns}$), the freshwater flux ($F_{wat}$) and the wind stress ( $\vec{\tau}$ ). The fluxes at the air-sea interface are computed within SURFEX taking into account near-surface atmospheric and oceanic parameters, following

a radiative scheme and a bulk parameterization.

$$Q_{sol} = (1-\alpha) SW_{down} \tag{2}$$

$$Q_{ns} = LW_{down} - \varepsilon \sigma T_s^4 - H - LE \tag{3}$$

$$F_{wat} = E - P_l - P_s \tag{4}$$

$$\vec{\tau} = \rho C_D (\vec{U}_s - \vec{U}_a)^2 \tag{5}$$

where $SW_{down}$ and $LW_{down}$ are the incoming short-wave (solar) and long-wave (infrared) radiative heat fluxes, respectively. H, LE and $\vec{\tau}$ are the sensible heat flux, the latent heat flux and the momentum flux (or wind stress), calculated thanks to a sea surface turbulent flux bulk parameterization. They depend on the wind speed and air-sea gradients of temperature, humidity and velocity, respectively. $\alpha$ is albedo, $\varepsilon$ is emissivity and $\sigma$ is the Stefan-Boltzman constant. Emissivity is a constant value, usually taken as 0.96 over ocean. Ocean albedo can be taken as a constant or can evolve following the Taylor

et al. (1996)'s formulation to account for the solar zenith angle, the Seferian et al. (2017)'s multi-spectral bands albedo accounting for solar zenith angle and wind speed, or the Salisbury et al. (2013)'s formulation.



E is total evaporation (included sublimation), $P_l$ and $P_s$ are liquid and solid precipitation in surface, respectively (directly coming from ATM component or from the atmospheric forcing in forced mode).

$\vec{U_a}$ is the wind at the first atmospheric level. $C_D$ is the drag coefficient calculated by the sea surface turbulent
fluxes parameterization. ρ is the air density.

$T_s$ and $\vec{U_s}$ are the ocean surface temperature and horizontal current. They are here the only oceanic parameters needed to compute the air-sea exchanges, and thus transferred from OCE to SURFEX. In return, SURFEX transfers the sea surface fluxes values $Q_{sol}$, $Q_{ns}$, $F_{wat}$ and $\vec{\tau}$ to the OCE component via OASIS3-MCT (Tab. 1).

Almost the same principles apply for the exchanges in the presence of sea-ice. In this case, SURFEX needs the sea-
ice cover ($C_{ice}$) from the ocean model and calculates fluxes as :

$$F_{ocean} = (1 - C_{ice})F_o + C_{ice}F_{ice} \tag{6}$$

where $F_o$ is the flux over open ocean calculated using the ocean properties and $F_{ice}$, the sea-ice flux, is calculated using the Charnock (1955)'s flux formulation. In this case, the radiative fluxes are calculated using the sea-ice temperature, albedo and emissivity. Emissivity is taken as constant over sea-ice whereas albedo and surface temperature are given by the ocean and
sea-ice model. In summary, in addition to the free ocean parameters usually needed, in the presence of sea-ice, SURFEX needs the sea-ice temperature, albedo and ice cover. In return, SURFEX transfers the area averaged sea-ice and open ocean momentum, heat and water flux to the OCE and ICE components respectively (Tab. 1). These mixed fluxes allow to conserve energy and water in the coupling. However, SURFEX also calculates and send pure sea-ice fluxes to the ICE component. These fluxes will be used in the ICE component to redistribute the total fluxes over ice-categories and open
ocean, but it will finally conserve the grid mean fluxes.

### 2.3.2 WAV-SURFEX-OCE

To represent the changes of sea surface roughness and ocean dynamics by surface gravity waves, the wave model WAVEWATCH III (WW3) (Tolman, 2002, 2009) has been coupled to SURFEX via OASIS3-MCT. WW3 solves the random phase spectral action density balance equation for selected wavenumber-direction spectra, parameterizing the effects of
wind-wave interactions, nonlinear wave-wave interactions, wave-bottom interactions, depth-induced breaking, dissipation, reflection of shorelines or floating objects such as icebergs. This model can be used for regional and global simulations on regular or unstructured grids.

SURFEX sends to WW3 (via OASIS3-MCT) the x and y components of the 10m-wind speed, as a direct forcing for the waves. In return, wave parameters like the significant height ($H_s$), the wave peak period ($T_p$), the Charnock parameter (α ;
Charnock, 1955) are sent from WW3 to SURFEX in order to take into account the sea state in the estimate of atmospheric surface fluxes.



The atmosphere momentum flux varies in function of the sea state through the drag coefficient $C_d$ depending on the roughness length $z_0$ as :

$$C_d = \frac{K^2}{\left( \log\left( \frac{z_a}{z_0} \right) - \Psi\left( \frac{z_a}{L} \right) \right)^2}$$

(7)

where $z_a$ is the height of the first atmospheric level, K the von Kàrmàn constant, $\psi$ an empirical stability function, and L the Monin-Obukhov length. The roughness length $z_0$ is related to the Charnock parameter $\alpha$ and to the friction velocity

$U_x$    through the Charnock's formulation (1955) with the smooth flow limit effect following Smith (1988) :

$$z_0 = \alpha \frac{u_x^2}{g} + 0.11 \frac{\nu}{u_x}$$

(8)

where $\nu$ is the kinematic viscosity of dry air and g the gravity acceleration.

Two different ways to represent the actual wave impact are available:

as in the ECMWF operational coupled IFS-WAM (Janssen et al., 2001), the Charnock parameter and roughness length are directly computed in WW3 as a function of the sea state, then transferred to SURFEX and used to compute the drag coefficient using eqs 7 and 8.

the wave parameters $H_s$ and $T_p$ are computed in the wave model, then transferred to SURFEX and used to compute the

Charnock parameter using the COARE 3.0 bulk flux algorithm (Fairall et al., 2003). These standard wave parameters present the advantage to be easily compared to observations. Two different wave-dependent approaches are available within COARE 3.0, making the roughness length dependent either on the peak period of the waves only (eq. 9 from Oost et al., 2002) or on both the peak period and the significant wave height (eq. 10 from Taylor and Yelland, 2001).

$$\alpha = 50 \left( c_p / U_x \right)^{-2.5}$$

(9)

$$z_0 = 1200 H_s \left( H_s / L_p \right)^{4.5}$$

(10)

where $c_p$ is the wave phase velocity and $L_p$ the wave length. In open ocean conditions, these two quantities can be related to the peak period using standard deep-water gravity wave relationships:

$$L_p = \frac{g T_P^2}{2\pi}; \quad c_p = \frac{g T_P}{2\pi};$$

Both approaches can be used in the SURFEX-WW3 coupling through choices of specific parameters in the

SURFEX namelist.

The exchanges between the ocean and the wave components (Fig. 1 (2b)) are directly managed by OASIS3-MCT and thus externalized from SURFEX. They are summarized here to fully describe the WAV-SURFEX-OCE coupled system



(See also Table 1). Surface waves induce Stokes drifts which impact the oceanic dynamics and the advection of the tracers. These are provided to the ocean model as the x and y components of the surface Stokes drift (USS) and the Stokes transport (TUS) (e.g. Breivik et al, 2014 ; Garnier et al, 2017) computed by WW3. The additional pressure associated to waves, namely the wave induced Bernoulli head pressure (BHD), is also provided by WW3 as well as the x and y components of the net wave-supported stress (TAW). This last term corrects the wind stress simulated by the atmospheric model from the part forcing the wave dynamics. The increase of ocean mixing due to wave breaking is represented through the wave-to-ocean turbulent kinetic energy flux (FOC). And at the sea surface, momentum from breaking waves (the x and y components of the wave ocean momentum flux, TWO) is a source to the ocean momentum. Likewise, the dissipation of the waves in the boundary layer increases ocean mixing (the corresponding energy flux due to bottom friction is FBB) and is a source to the ocean momentum (represented by the x and y components of the momentum flux due to bottom friction, TBB). Lastly, the increase of the bottom friction due to the wave presence is represented as a function of the current, the root mean square amplitude of the orbital velocity induced by the waves (UBR). Finally, the significant wave height (Hs) is also used to define the vertical extent of the wave effects. All the variables are sent from WW3 to the ocean model through OASIS. In return, the ocean model sends to WW3 (Fig. 1 (2b) - OCE to WAV) the sea surface height (SSH) and the surface currents (SSU, SSV for the x and y components).

### 2.3.3 HYD-SURFEX-OCE

Originally, several embedded couplings of SURFEX with hydrological models were developed for the CNRM version of the Total Runoff Integrating Pathways (CTRIP) River Routing Model at the global scale (Decharme et al. 2010), the MODCOU hydrogeological model over France (Habets et al., 2008), and the TOPography based MODEL (TOPMODEL) hydrological model at the meso-scale (Bouilloud et al., 2009), respectively. However, in order to ease the upgrade of the hydrological model in these coupled configurations in the future, it was decided to implement the coupling with an hydrological model in the standard coupling interface, using the CTRIP hydrological model as a reference/example for this development.

Hydrological models are used to compute three processes: groundwater dynamic, river runoffs and discharges to the ocean. Some of the hydrological models simulate also floodplains as the CTRIP model that represents the large seasonal floodplains. These processes are strongly coupled to the land surface water and energy budget, and feedbacks to the ocean. for this purpose, the CTRIP river routing model has been coupled to SURFEX via OASIS3-MCT.

In SURFEX, the continental water and energy budgets are computed using the ISBA scheme. The soil moisture and heat vertical transports are explicitly solved using a multi-layer scheme (Boone et al., 2000; Decharme et al. 2011). This scheme has been validated over many local field datasets (Boone et al., 2000; Habets et al., 2003; Decharme et al. 2011) or regional studies (Decharme et al. 2013, 2016), improving confidence in the model's ability to consistently represent a variety of environmental conditions in different climate regimes. It includes a comprehensive sub-grid hydrology to account for the



heterogeneity of precipitation, topography and vegetation in each grid cell (Decharme and Douville 2006; Decharme et al 2013).

SURFEX sends to CTRIP the surface runoff ($R_{nf}$), and the deep soil drainage ($D_r$) calculated in ISBA, as well as the freshwater flux over floodplains open water to the atmosphere ($F_{WFP}$, Decharme et al. 2012). The deep soil drainage represents the infiltration of water that is directly routed to the ocean in the absence of groundwater in the grid cell.

Concerning the groundwater, CTRIP in turn sends to SURFEX the water table depth (*WTD*) of the groundwater, i.e. the depth of the groundwater, and the grid-cell groundwater fraction ($C_{WTD}$). In other words, the *WTD* computed in CTRIP acts as the lower boundary condition for the ISBA soil moisture diffusive equation (Vergnes et al. 2014). For the floodplains, CTRIP sends to SURFEX the floodplain grid-cell fraction ($C_{FP}$) and the floodplains water mass flux to the land surface reservoir ($W_{FP}$). Finally CTRIP sends to the oceanic model the discharges at the mouth of all rivers ($D_{is}$).

## 3 Multi-model and multi-scale applications

This section illustrates various applications of the standard coupling interface.

### 3.1 The global climate configuration CNRM-CM6

The CNRM-CM climate coupled model is designed to perform global atmosphere-ocean coupled integrations over centuries to millennium. Its former version CNRM-CM5 (Voldoire et al., 2013) has been used to perform the 5[th] phase of the 280 coupled model intercomparison panel (CMIP). The model was coupled with the OASIS3 coupler but the coupling interface was embedded in the atmospheric model, not directly in SURFEX. The newly designed model, CNRM-CM6 combines all the components pictured on Figure 1, except WAV. It is based on the ARPEGE-Climat (v6) atmospheric model at about 140km resolution, the NEMO (v3.6) ocean model (Madec et al., 2008) and the GELATO (v6) sea-ice model both at a 1° nominal resolution, and the CTRIP river routing model at 0.5°. Figure 2a shows the orography and bathymetry of the 285 atmosphere and ocean models. SURFEX v8 is embedded in ARPEGE-Climat and activates the FLAKE scheme over lakes and the ECUME v6 flux parameterization over ocean (Belamari, 2005). All components are coupled through OASIS3-MCT every hour. Before fully assembling the model, the different components are tested in more constrained configurations. Here we illustrate how the new SURFEX coupling interface allows to analyse the coupling of the different components step by step.

### 3.1.1 Land Surface Hydrology (SURFEX-CTRIP)


First, we assess the performance of the land surface model alone driven by the atmospheric forcing provided by Princeton University (http://hydrology.princeton.edu) at a 3-hourly time step and 1° resolution. This data set is derived from the National Center of Environmental Prediction-National Center for Atmospheric Research (NCEP-NCAR) reanalysis with





corrections on systematic biases and precipitation disaggregation. More details about this forcing product can be found in
Sheffield et al. (2006). In this context, the model is run over the period 1948-2010 at the forcing resolution (1°). Figure 3a,
shows the summer mean land surface evaporation averaged over the period (1980-2009). The zonal mean evaporation
compares relatively well with the observed estimate from Jung et al. (2009).

In a second step, we assess the performance of the whole continental hydrologic system by coupling SURFEX with
the CTRIP river model with the same atmospheric forcing. As described in section 2.2.3 the surface and groundwater
reservoirs exchange water with the rivers. From figure 3b, it can be seen that the coupling does not change the realism of the
evaporation flux, despite the increased level of freedom.

### 3.1.2 The global climate model : closed water cycle (ARPEGE Climat-SURFEX-CTRIP-NEMO-GELATO)

As a third step, the full land surface hydrology system is run online with the atmospheric model. In this context,
SURFEX is used at the atmospheric model resolution (~140km) over the period 1978-2010. The land surface evaporation
flux is impacted by the coupling with the atmospheric model (figure 3c): the high latitude evaporation is overestimated due
the radiative and precipitation biases of the atmospheric model. However, the global evaporation patterns are realistic.

Finally, the full system CNRM-CM6 has been integrated over the period 1950-2010. In the full system, the water
cycle is fully dynamic and any unbalance will be reflected by a sea surface height drift. In the full system, the simulated land
evaporation realism is similar to the atmospheric simulation (figure 3d to compare with figure 3c), and the sea surface height
drift is +0.2m/century which is a good performance compared to CMIP5 models.

### 3.2 CNRM-RCSM6 over the Mediterranean Sea (ALADIN Climat-NEMOMED12-CTRIP)

Regional Climate System Models (RCSM) belong to the same family as the global earth system models (GCM)
used in the CMIP experiments, but generally have a higher resolution over a limited-area domain. The new CNRM-RCSM6
version presented here is the limited area counterpart of CNRM-CM6. As such it represents the same components: ATM,
OCE and HYD. The atmosphere model ALADIN-Climate v6 is the regional version of ARPEGE-Climate v6 presented with
the CNRM-CM6 configuration, and is linked with the same SURFEX v8 version. The resolution of the atmosphere grid is
about 12 km. The CTRIP model presented in 2.2.3 is also included, with a 0.5° resolution. Finally, the ocean model is
NEMOMED12, with a resolution about 6 km (1/12° grid) (Beuvier et al., 2012, Hamon et al., 2016).

Only the CTRIP cells included in the Mediterranean and Black Sea drainage areas are considered ; for the Nile river
basin, the runoff is prescribed by a 12-month climatology. The NEMOMED12 ocean model doesn't consider the Black Sea
in its domain, and the latter is considered as a river inducing the difference of salinity between the Aegean Sea and the
Marmara Sea. In the coupled system, the river discharges of the Black Sea drainage area (58 river mouths in the CTRIP 0.5°
grid) are summed with the Evaporation minus Precipitation (E-P) budget from SURFEX over the Black Sea. Figure 2b





shows the land-sea mask and the orography of ALADIN-Climate and the bathymetry of NEMOMED12, with the drainage
areas of the Black Sea and of the Mediterranean Sea in the CTRIP 0.5° grid. Outside the domain of NEMOMED12, in the
Atlantic Ocean, the Baltic Sea and the Black Sea, an SST field must also be provided to the atmospheric model; it is usually
coming from the same dataset as the lateral boundary conditions supplied to the regional atmospheric model.

    The atmosphere-ocean coupling frequency is 1 hour, so that the diurnal cycle of the ocean Sea Surface Temperature
(SST) can be simulated. To assess the realism of the SST diurnal cycle simulated by the model under present day conditions,
the model has been run using reanalysis data at the lateral boundaries for the atmosphere and ocean components. Spectral
nudging is applied in addition to the ERA-Interim reanalysis over the inner model domain (Berrisford et al., 2009). ORAS4
is used as Atlantic boundary condition for the ocean model (Balmaseda et al., 2013). Additionally, the solar penetration on
the upper ocean is prescribed using monthly mean maps of chlorophyll-a concentration (Ocean Colour Climate Change
Initiative dataset, European Space Agency, available online at http://www.esa-oceancolour-cci.org/) .

In Figure 4, we compare the amplitude of the diurnal cycle of the SST simulated by the model to the Météo-France
Lion buoy observations, on the 2009-2013 period (the model has been running since 1979 with a 24h-coupling frequency
before this comparison). The amplitude of the diurnal cycle of the SST is computed as the difference between the maximum
of the hourly mean SST between 09UT and 17UT, minus the minimum between 18UT the day before and 8UT. As a first
estimation, we consider that we can compare the SST of the buoy (at 1 m depth) to the SST of the model (temperature of the
1-m thick 1st layer). We use data only for days without any missing value in the observations, and the model value is taken at
the grid point nearest to the buoy. Only amplitudes above 0.1°C are kept, and the results are split according to the season.
The means, standard deviations and correlations are computed on the two series composed by the amplitudes on the same
days for both dataset.

    In average the model simulates realistic amplitudes of the diurnal cycle of the SST, though slightly overestimated in
spring (AMJ, +0.2°C) and summer (JAS, +0.3°C) and underestimated in winter (OND, -0.3°C). The amplitudes are better
correlated in winter and spring (correlation equal to 0.6 and 0.7). In summer, the model doesn't reproduce the extreme values
up to 4.5°C. In autumn, it also underestimates the high values. For these two seasons the correlation is equal to 0.4 and 0.3
respectively. The standard deviations are between 0.2 and 0.7 according to the season, while they are between 0.3 and 0.6 for
the buoy.

Considering that the model grid-point cannot exactly reproduce the local buoy SST, these results show that CNRM-
RCSM6 represents a realistic diurnal cycle of the SST, and they allow to enlarge the study of the links between air-sea fluxes
representation and diurnal cycle.

### 3.3 AROME-NEMO WMED

    The AROME-NEMO coupled system combines the operational kilometer-scale non-hydrostatic Numerical Weather
Prediction (NWP) system of Météo-France, AROME (Seity et al., 2011), and the NEMO ocean model (Madec et al., 2008).





There is no ICE, no WAV and no HYD component. The AROME-NEMO coupled system aims at better representing the ocean-atmosphere coupled processes at fine-scale and at assessing the impact of the coupling on short-range forecast of severe weather.

The AROME-NEMO system is first applied over the western Mediterranean Sea (WMED) (Fig. 2c). It involves a 2.5km-resolution configuration of AROME, named AROME-WMED (Fourrié et al., 2015), in version cy38t1 as atmospheric component model. AROME-WMED was designed and operated in real time during the field campaigns of the HyMeX program (Drobinski et al., 2014; Ducrocq et al., 2014; Estournel et al., 2016a). The surface scheme in AROME-WMED is SURFEX v7.2. The OCE component model is NEMO version 3.2 with a domain covering the western Mediterranean Sea at a horizontal resolution of $1/36°$ (an ORCA grid with resolution between 2.2 and 2.5 km). This configuration, named
WMED36 is described in Lebeaupin Brossier et al. (2014).

The impact of the coupling is illustrated here for the Intense Observation Periods (IOPs) 13 and 16b from the first HyMeX field campaign (11-15 October 2012, 25-29 October 2012; Ducrocq et al., 2014; Rainaud et al., 2016). The coupled experiment is called CPL. For the ATM component, the initial conditions come each day from the AROME-WMED analysis at 00UTC and the lateral boundary conditions come from the 10km-resolution ARPEGE operational forecasts with a hourly
frequency. The ocean initial conditions on 11 and 25 October come from a former ocean-only NEMO-WMED36 simulation, driven in surface by the AROME-WMED operational forecasts for range $t_0+1h$ to $t_0+24h$ since the $1^{st}$ September 2012. The ocean open-boundary conditions are the monthly-averaged PSY2V4R4 (Lellouche et al., 2013) analysis provided by Mercator Océan.

CPL runs for 48h-range each day starting at 00UT. OCE is restarted each time from the previous run ($t_0+24h$ of the
previous day). The coupling frequency is 1 hour and the interpolation method is bilinear. The Atlantic Ocean, the Adriatic Sea and the western Ionian Sea are uncoupled. In these areas (grey marine zones in Fig. 2c), SST comes from the SURFEX initial state and remains constant, and horizontal current is considered as null. The CPL experiment is compared to a atmosphere-only simulation, called ARO, with the same atmosphere and SST initial fields as CPL. But, the SST field does not evolve during the forecast run in ARO.

Figure 5 illustrates the coupling impact at the Lion buoy location (4.7°E-42.1°N) along the 48h forecasts starting at 00 UT on 12 and 13 October (Fig. 5a, IOP13) and on 26 and 27 October (Fig. 5b, IOP16b) IOP13 is characterized by moderate northerly wind till early morning on 13 Oct., followed by southerly moist wind and precipitation on 14 Oct. IOP16b is characterized by strong northerly cold winds from 27 Oct. around noon. The time evolution of SST in CPL well follows the observed evolution. Indeed, CPL represents the sea surface cooling in response to the abrupt wind speed increase
on 27 Oct afternoon. The SST decrease during the CPL forecast run is of about 2.5°C to compare to 3.5°C in the observations (Fig. 5b). The SST from ARO is constant during all the forecast runs and thus misses this rapid sea surface cooling. As a consequence, the latent heat flux (LE) from ARO is larger by more than 150 W/m$^2$ (~15%) than the CPL one on 28 Oct. The differences for IOP13 are weaker, by 16W/m$^2$ on average (~5%). Some differences are also seen for the





precipitation rate on 14 Oct and on 26 Oct. The low-level wind and temperature are close in the two experiments and in
agreement with observations, especially for short-range (0-24h) forecasts.

As illustrated here, the impact of the interactive coupled ocean can be significant for intense weather situations with abrupt changes in the wind speed and/or in the surface fluxes during the forecast, especially for longer term forecasts (24-48h) (see Rainaud et al, in revision). Further investigations are ongoing to evaluate the coupled processes impact on other case studies.

**3.4 MESONH-MARS3D-WAVEWATCH III**

The Meso-NH-MARS3D-WW3 (M2W) model system couples the Meso-NH atmospheric model (Lafore et al. 1998), the MARS3D (Model for Application at Regional Scale) oceanic model (Lazure and Dumas 2008) and the WW3 (WAVEWATCH III) wave model (Tolman, 2002, 2009). This system, coupling limited-area atmospheric, oceanic and wave models, aims at better representing and understanding the ocean-wave-atmosphere interactions at very fine scales (from
100m to few km horizontal resolution). The M2W system has been applied over the Iroise sea, a good location to study the interactions between ocean, wave and atmosphere. The Iroise sea indeed exhibits two marked characteristics, namely a strong tidal current named Fromveur with an intensity up to 2m/s and an intense SST seasonal front from April to October named the Ushant front. In this area waves come from a large fetch.

In the present study, the horizontal resolution of Meso-NH is 2 km with 80 grid points in each direction and the
vertical resolution goes from 10 m near the surface to 700m at the model top (25 km) using 100 grid points. MARS3D is run with 40 vertical generalized-sigma levels (over bottom depths lower than 200m) and horizontal resolution of 500m with 328 and 324 points in the longitude and latitude directions, respectively. This oceanic model configuration is extracted from the AGRIF zoom embedded into the Bay of Biscay configuration (2.5 km) of the coastal operational oceanography system Prévimer (Lazure et al., 2009). The WW3 model is used with a resolution of about 1.5km in each direction with 103 points
for the latitude and 119 points for the longitude. The spectral resolution corresponds to 32 frequencies chosen from 0.0373 to 0.7159 Hz and 24 points for the propagation direction (every 15 degrees). For a complete description of the WW3 configuration, the reader can refer to Boudière et al. (2013).

The simulation was performed on a 24h period starting the 2nd September 2011 at 00 UTC. The three models have different resolutions and time steps (10s for Meso-NH, 20s for MARS3D and 10s for all parts of WW3) and cover almost the
same spatial area (fig 2d). Exchanged fields are spatially extended over a few cells when required. The coupling frequency between the three models is 100s and the method of spatial interpolation is a distance weighted nearest-neighbour interpolation.

The initial and forcing fields for Meso-NH are computed from Meteo-France operational analysis of AROME available every 3h. At open boundaries, WW3 is forced by 3-hourly energy spectra from HOMERE hindcast database
(Boudière et al., 2013) provided by IFREMER. The initial and open boundary conditions for MARS3D are extracted from the hourly outputs of the Iroise AGRIF zoom embedded into the Prévimer Bay of Biscay configuration. For a suitable tide



representation, the sea surface and barotropic currents have been previously saved at every time step and used in the open boundary conditions that force the present offline configuration of the Iroise zoom.

Figure 6a shows, for the M2W coupled run, the wind stress as computed in SURFEX and the SST (white solid contours) on September 2nd 2011 at 9h, 2 hours after the high tide. For the present situation with moderate winds, the wind stress is largely driven by the meso-scale and submeso-scale ocean dynamics of the Ushant front, realistically simulated by the high resolution MARS3D model. Indeed, the pattern of the wind stress follows the SST. The wind stress weakens when air is blowing from warm to cold SST region. This strong dependence to oceanic mesoscale is slightly modulated by the atmospheric-ocean feedbacks: wind stress as simulated by the fully coupled ocean-atmosphere-wave dynamics and the ocean-atmosphere retroaction alone exhibits larger differences where the wind is stronger and along the Ushant thermal front (Fig. 6c). As expected, the presence of islands has a clear impact. In all experiments, the atmospheric roughness length is estimated at the air-sea interface by Eq. 8. The Charnock parameter is directly calculated by the wave model for the M2W system and estimated from the wind in the other model configurations. In M2W simulations, the roughness length (Fig. 6b) is larger over the warm part of thermal front where the wind is maintaining a young wind sea and above cold oceanic water in the Channel. Similitudes between the patterns on figures 6c and 6d confirm that the wind sea representation impacts the atmospheric dynamics through the wind stress. However, we have to keep in mind that the impact is sensitive to the choice of the representation of the sea state (not shown), which remains an open scientific question.

The fully M2W system allows a continuous adjustment between the atmosphere, the ocean and the waves dynamics. In particular, the tidal oscillation of the Ushant front location and the wind sea state are taken into account in the atmosphere evolution, with a coherent retroaction between the 3 components. However, the physical representation of some key parameters at the air-sea interface requires further investigations (e.g. parameterization of the wave effect into the surface atmospheric boundary layer). The standard coupling interface has simplified the technical development of M2W. Beyond the technical phase, more detailed analysis are underway to determine the importance of the dynamical coupling with respect to forcing and to establish the necessary spatial and time resolution as well as the coupling frequency to properly represent the dynamical and thermodynamical couplings.

### 3.5 MESONH-SYMPHONIE

The MESONH-SYMPHONIE coupled system combines the non-hydrostatic mesoscale atmospheric model of the French research community, Meso-NH (Lafore et al, 1998), and the regional and coastal ocean model SYMPHONIE (Marsaleix et al., 2008, 2009, 2012). In this application, there is no ICE, no WAV and no HYD component. Surface turbulent fluxes are calculated with SURFEX v7.2 embedded in the Meso-NH version 5.1.2. This coupled system has been developed to focus on regional and coastal issues in the Mediterranean Sea, including dense ocean water formation (Marshall and Schott, 1999, Estournel et al., 2016b), coastal and frontal dynamics, and Heavy Precipitations Events (HPE) (Ducrocq et al., 2014). The HyMeX program (Drobinski et al., 2014, Ducrocq et al., 2014, Estournel et al., 2016a) provides an ideal framework for studying these processes.





This study focuses on the IOP16 of the HyMeX program (October 25, 2012 to October 29, 2012, Ducrocq et al, 2014; Lebeaupin Brossier et al, 2014; Rainaud et al., 2016). IOP16 can be separated into two parts: IOP16a (25 to 27 October) dedicated to HPE and IOP16b (27 to 29 October) dedicated to Strong Wind Events (SWE). The purpose of this study is to show the impact of the air-sea coupling on the HPE and the reduction of oceanic stratification during SWE, on the basis of coupled and uncoupled simulations.

Two coupled systems are used with different resolution: a 10km resolution (hereafter low-resolution) convective parametrized atmospheric model, MESONH-10km-SYMPHONIE-1km, and a 2.5km resolution (hereafter high-resolution) convective permitting atmospheric model, MESONH-2.5km-SYMPHONIE-1km. In both cases the horizontal resolution of the ocean model is 1km. The spatial extension of the atmospheric numerical domain is the same at both resolutions and is wider than the oceanic domain (Fig. 2e). In uncoupled zones (grey marine zones in Fig. 2e), fluxes are calculated with the

OSTIA SST analysis (Donlon et al., 2012) with a spatial resolution of 6km. The initial state and boundary conditions of the ocean model are provided by the fields of the MERCATOR-OCEAN PSY2V4R4 analysis (Maraldi et al., 2013) corrected with in-situ CTD profiles collected in the Northwestern Mediterranean Sea during summer 2012 and 2013 (Estournel et al., 2016b). The boundary conditions of the low and high resolution atmospheric model use the analyzes produced by the ECMWF meteorological center with a sampling of 6 hours and a horizontal resolution of 1/8 °. The low resolution

atmospheric model is initialized from the same analysis whereas the high resolution atmospheric model is initialized from the low resolution. In the following, only the results of the high resolution model are shown.

Two high-resolution simulations of the IOP16 were performed. The first simulation is a coupled simulation (CPL) using a coupling frequency of 600 seconds and a bilinear interpolation method. The second simulation is no longer coupled (UNCPL), in the sense that the ocean model no longer sends its surface temperature to the atmospheric model. Air-sea fluxes

are actually calculated with the OSTIA SST and provided to the ocean model using the same frequency and interpolation method as the coupled simulation. The high-resolution simulations (UNCPL and CPL) begin on October 25 at 00UT and end on October 29 at 00UT. The initial state is provided by the interpolation of a low-resolution simulation started on August 2, 2012.

Figure 7 shows the non solar net heat fluxes (Qns) and the SST (black contours), averaged over the two parts of the

IOP16, for UNCPL and CPL simulations. In the UNCPL simulation, the OSTIA SST does not contain the meso-scale and sub-meso-scale oceanic structures (Fronts, Eddy, Filaments), unlike in the case of the CPL SST. These oceanic structures influence the spatial distribution of fluxes. Furthermore, during IOP16b, intense fluxes are generated by the SWE. In the CPL simulation, the SST is reduced by the feedback of these fluxes, while in the uncoupled simulation the fluxes rely on the slowly varying (since daily averaged) prescribed OSTIA SST. These differences on the fluxes have an impact on the HPE

during IOP16a and the oceanic stratification loss during IOP16b.

In summary, the ocean-atmosphere coupling has an impact on the HPE and the oceanic stratification. Further investigations are ongoing on this case-study and other HyMeX case-studies to better understand coupled processes. Furthermore, the coupling impact at different scales could be evaluated.



### 3.6 MESONH-NEMO Indian Ocean

The MESONH-NEMO coupled system combines the Meso-NH model version 5.4.1 (Lafore et al, 1998) and the NEMO v3.6 ocean model (Madec et al., 2008). The SURFEX version used for this system is SURFEX v7.3. There is no ICE, no WAV and no HYD component. The MESONH-NEMO coupled system aims at better understanding and representing the ocean-atmosphere coupled processes at meso-scale, with a particular focus on their role on tropical cyclone development and air-sea fluxes in extreme wind conditions. This study focuses on the simulation of the tropical cyclone BEJISA, which passed

very close to La Reunion island between January 1st and January 3rd of 2014.

The MESONH-NEMO system is first implemented over the South West Indian Ocean (SWIO). The horizontal resolution of MESONH is 8 km with 150x125 grid points, and 72 vertical levels up to 25 km height. The OCE component model is NEMO in a configuration over the SWIO with a horizontal resolution of 1/12° over an ORCA grid (i.e. around 9km-resolution) (Figure 2f). Initial atmospheric conditions are taken from AROME-INDIEN (a research version of AROME

over the SWIO which includes 3DVAR data assimilation and a 1D ocean mixed layer), atmospheric boundary conditions are taken from ECMWF analysis, while initial and boundary oceanic conditions are given by the PSY4V2R2 analysis from CMEMS (E.U. Copernicus Marine Environment Monitoring Service). Two simulations starting on January 1st 2014 and for a 36h period are performed: a reference simulation, called NOCPL, in which only the atmospheric model is run and a second one called CPL, in which the coupled system is run with a 1h coupling frequency. The simulated trajectory of Bejisa is quite

similar in both simulations and very close to the best track at all times (not shown). This is consistent with the fact that the tropical cyclone trajectory is mainly driven by the large-scale dynamics.

Figure 8 shows the integrated total water content (ITWC in mm) after 18, 24 and 30 hours of simulations for both simulations. This figure illustrates that the atmosphere-ocean coupling has a strong impact on the microphysical structure of the tropical cyclone. After 30h of simulation, both the region of maximum ITWC and its intensity are different. By looking at

each hydrometeor individually (not shown), one can see that the difference is mainly due to the graupel distribution. This is still an ongoing work, the next step will be to increase the horizontal resolution for the atmosphere to resolve explicitly the microphysics and study into more details the impact of the oceanic coupling on the microphysical structure of tropical cyclone.

## 4 Conclusions and perspectives

The SURFEX platform is shared by many atmospheric models to consider various surface properties and provide the surface fluxes of heat, water, momentum and carbon to the atmosphere. This platform is used for a wide range of applications from academic simulations with the MESO-NH model to numerical weather forecasting with the AROME




model and to climate projections with the ARPEGE GCM. This platform is also used in a standalone mode driven by various atmospheric forcings. It is used for applications over a wide range of spatial scales from local sites to global climate model
scales (~150km). A quite generic coupling interface using OASIS3-MCT has been implemented in SURFEX to allow the coupling of this surface model to other models such as hydrological models, ocean models and wave models. This standard interface has been successfully used for diverse purposes with several atmospheric and ocean models, as shown in part 3.

One of the strength of this approach is that all atmospheric models using SURFEX share the ability of coupling with ocean/wave/hydrological models. This means that any future development will be readily available to all SURFEX
users. Contrary to what could be expected, such a coupling is computationally efficient since the computational cost of the OASIS3-MCT interface is negligible compared to the cost of the models involved (as long as post-processing global conservation is not activated in OASIS3-MCT). Additionally, the computational cost of atmospheric component represents more than 55% of the total coupled model cost in all applications presented here. This ratio is even greater than 80% for CNRM-CM, CNRM-RCSM, AROME-NEMO and MESO-NH-Symphonie, meaning that the computational cost of
atmospheric applications is not severely impacted by the coupling in these cases. This is a crucial point in the perspective of high-resolution coupled numerical weather prediction systems.

The standard coupling interface with OASIS3-MCT is part of the version 8.0 of SURFEX, for the ocean/sea-ice/hydrology coupling. Soon, a next version will add the coupling with the wave model. Other improvements are planned in the near future. First, new coupling fields should be introduced, such as ocean salinity which has an impact on water fluxes,
marine aerosols emission and carbon concentration to close the global carbon cycle. The way currents interact with the momentum budget has also to be improved. SURFEX could also be used in standalone mode coupled to an ocean model. Ocean model are usually run in forced mode either forced by fluxes or forced by near surface atmospheric fields. In the last case, bulk formulae are embedded in the ocean code to calculate turbulent fluxes. The comparison with coupled run is unfair in this case as bulk formulae used in the ocean forced system are not the same as those used in the coupled system. Using
SURFEX on top of the ocean model in forced mode would resolve this issue as the fluxes would be calculated in SURFEX with the exact method used in coupled mode. However, in its current version, SURFEX can not be used over fully generic curvilinear grids as used in ocean models. To avoid the interpolation and allow SURFEX to exchange directly with the ocean in offline mode, it is necessary to support such grids in SURFEX. A configuration SURFEX-OCE-HYD also allows to run all the surface hydrological water cycle forced by an atmospheric observed forcing.

It is often hard to clearly isolate the pure coupling effect in existing applications, and it is not the purpose of this paper. In most cases, comparison of coupled and uncoupled simulations rather show the benefit of using a more detailed SST field in term of space and time variations than the pure coupling effect. Nevertheless, for operational purpose, coupled models appear as very promising tools able to represent and to take into account the rapid SST evolution in coherence (balance) with the atmosphere. It actually compensates the general lack of observations of the sea surface, even greater in
severe weather situations, related to the too few in-situ data over the ocean and to the larger amount of missing satellite observations in cloudy situation.



Similarly for climate studies, long term simulations have to be run including the ocean and sea-ice components. The main question is rather to assess that model performances are not greatly altered by the coupling having in mind that introducing the coupling is a step towards a more physical representation of the mixed layers both in the ocean and

atmospheric systems.

The six coupled systems presented here show a wide panel of applications using SURFEX and OASIS3-MCT, in term of spatial resolution (from 500 m in the ocean and 2.5 km in the atmosphere, to 1° and 140 km, respectively), and in term of scientific objectives: climate projections (CNRM-CM6), regional climate studies (CNRM-RCSM6), weather forecasts (AROME-NEMO WMED), ocean-atmosphere interaction and extreme events (MESONH-Symphonie and

MESONH-NEMO Indian Ocean), and very fine scale processes (MESONH-MARS3D-WW3) .

Sharing a standard coupling interface favours the scientific collaboration between SURFEX users. In particular, there is a need to work on coupled model initialisation techniques. In atmospheric and oceanic models, assimilation methods are on a rather mature stage, whereas this a new research field for coupled models (Laloyaux et al., 2016).

**Code availability**

The OASIS-SURFEX interface is available in the version 8 of SURFEX (http://www.cnrm-game-meteo.fr/surfex). The SURFEX code is freely available (Open-SURFEX) using a CECILL-C Licence (a French equivalent to the L-GPL licence; http://www.cecill.info/licences/Licence_CeCILL-C_V1-en.txt), excepted for the GAUSSIAN grid, the LFI and FA I/O formats, the dr HOOK tool.

OASIS3-MCT can be downloaded at https://verc.enes.org/oasis/download . The public may copy, distribute, use, prepare

derivative works and publicly display OASIS3-MCT under the terms of the Lesser GNU General Public License (LGPL) as published by the Free Software Foundation, provided that this notice and any statement of authorship are reproduced on all copies

The NEMO model can be downloaded at http://www.nemo-ocean.eu/ after a user registration on the NEMO website, NEMOMED12 configuration is available on demand to thomas.arsouze@ensta-paristech.fr, NEMO-WMED36 is available

on demand to cindy.lebeaupin-brossier@meteo.fr.

The SYMPHONIE model can be downloaded at http://sirocco.omp.obs-mip.fr/ after a user registration on demand to sirocco@aero.obs-mip.fr

The use of MARS3D model requires a license agreement (contact on the website). Once registered on the MARS3D website, users can access to the USHANT configuration on demand to valerie.garnier@ifremer.fr .

Meso-NH is freely available under CeCILL-C licence agreement. Version 5.3 includes SURFEX v8_0 and thus the coupling interface with OASIS3-MCT (for ocean/sea-ice/hydrology). Meso-NH can be downloaded at http://mesonh.aero.obs-mip.fr/mesonh53/





WW3 is distributed under an open source style license through a password protected distribution site at

http://polar.ncep.noaa.gov/waves/wavewatch/

ARPEGE/ALADIN/AROME are not available in open source. ARPEGE-Climate is available to registered users for research purposes only.

Outputs from all models discussed here are available on demand upon request to the authors.

### Acknowledgements

The authors acknowledge the MISTRALS/HyMeX program and the ANR-2012-BS06-0003 ASICS-Med funding. The
authors acknowledge the DGA (Direction Générale de l'Armement), a part of the French Ministry of Defense, for its

contribution to Romain Rainaud's PhD, as well as the coastal part of the Copernicus project co-funded by the MEDDE

(French Ministry of Ecology, Sustainable Development and Energy) and INSU (National Institute of Sciences and Universe).

The authors acknowledge Edmée Durand and Yann Drillet from Mercator Océan technical support. Simulations of the

MESONH-SYMPHONIE coupled system were performed using HPC resources from CALMIP (grants P09115 and P1325)

### 595 Author contribution

A.Alias: ARPEGE-Climate/ALADIN-Climate, and their SURFEX interface code management and tests.

S. Bielli: MESONH-NEMO coupled system development and validation in the SWIO

B. Decharme: SURFEX-OASIS interface coding, CTRIP-OASIS interface coding, and climate hydrological and global

model applications.

V. Ducrocq: Coordination/ Investigator of the SURFEX-OASIS interface development team.

S. Faroux: SURFEX code management.

C. Lebeaupin Brossier: SURFEX-OASIS interface adaptation to limited-area models and verification; AROME-NEMO

WMED coupled system development.

F. Léger, R. Rainaud, H. Giordani, S. Riette: AROME-NEMO WMED application and validation.

J. Pianezze, V. Garnier, M.-N. Bouin, J.-L. Redelsperger, M. Accensi, F. Ardhuin : development and application of the ocean-

atmosphere-wave coupling.

L.Seyfried, P. Marsaleix, E. Richard MESONH-Symphonie coupled system development and validation.

F. Sevault: ALADIN-NEMOMED12-CTRIP development and validation.

S.Valcke: OASIS3-MCT development.

A. Voldoire: SURFEX-OASIS interface technical development and tests in the global climate model.



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





## Tables

| Annotation | SOURCE model to TARGET model field description | Name in SURFEX namelist |
|---|---|---|
| (1a) | **OCE to SURFEX** | |
| $T_s$ | Sea surface temperature | CSEA_SST |
| $U_s$ | Sea surface zonal current | CSEA_UCU |
| $V_s$ | Sea surface meridian current | CSEA_VCU |
| | **SURFEX to OCE** | |
| $\tau_u$ | Wind stress zonal component | CSEA_FWSU |
| $\tau_v$ | Wind stress meridian component | CSEA_FWSV |
| $Q_{ns}$ | Non solar net heat flux | CSEA_HEAT |
| $Q_{sol}$ | Solar net heat flux | CSEA_SNET |
| $\tau$ | Near surface wind speed | CSEA_WIND |
| $U$ | Wind stress module | CSEA_FWSM |
| $E$ | Evaporation | CSEA_EVAP |
| $P_l$ | Liquid precipitation | CSEA_RAIN |
| $P_s$ | Solid precipitation | CSEA_SNOW |
| $F_{wat}$ | Net water flux (Eq. 4) | CSEA_WATF |
| $P_{surf}$ | Surface Pressure | CSEA_PRES |
| (1b) | **ICE to SURFEX** | |
| $T_{sice}$ | Sea-ice temperature | CSEAICE_SIT |
| $C_{ice}$ | Sea-ice cover | CSEAICE_CVR |
| $\alpha_{ice}$ | Sea-ice albedo | CSEAICE_ALB |
| | **SURFEX to ICE** | |
| $Q_{ns}$ | Non solar net heat flux over sea-ice | CSEAICE_HEAT |
| $Q_{sol}$ | Solar net heat flux over sea-ice | CSEAICE_SNET |
| $E_s$ | Sublimation | CSEAICE_EVAP |
| (2a) | **WAV to SURFEX** | |
| $H_s$ | Significant wave height | CWAVE_HS |
| $T_p$ | Peak period | CWAVE_TP |





| | | |
|---|---|---|
| $\Box$ | Charnock coefficient | CWAVE_CHA |
| $U_s$ | Sea surface zonal current | CWAVE_UCU |
| $V_s$ | Sea surface meridian current | CWAVE_VCU |
| | **SURFEX to WAV** | |
| U10 | 10m zonal wind speed | CWAVE_U10 |
| V10 | 10m meridian wind speed | CWAVE_V10 |
| (2b) | **WAV to OCE** | |
| HS | Significant wave height | - |
| USS | x and y component of the surface Stokes drift | - |
| TUS | x and y component of the Stokes transport | - |
| BHD | wave induced Bernoulli head pressure | |
| TAW | x and y component of the net wave supported stress | - |
| TWO | x and y component of the wave ocean momentum flux | - |
| FOC | Wave to ocean turbulent kinetic energy flux | - |
| TBB | x and y components of the wave boundary layer momentum flux | - |
| FBB | wave boundary layer turbulent kinetic energy flux | - |
| UBR | Root mean square amplitude of the orbital velocity of the waves | - |
| | **OCE to WAV** | |
| SSH | Sea surface height | - |
| $U_s$ | Sea surface zonal current | - |
| $V_s$ | Sea surface meridian current | - |
| (3a) | **HYD to SURFEX** | |
| WTD | Water table depth | CWTD |
| $C_{WTD}$ | Grid-cell fraction of water rise | CFWTD |
| $C_{FP}$ | Flood plains fraction | CFFLOOD |
| $W_{FP}$ | Flood plains potential infiltration | CPIFLOOD |
| | **SURFEX to HYD** | |
| $R_{nf}$ | Surface runoff | CRUNOFF |
| $D_r$ | Deep drainage | CDRAIN |
| $F_{CALV}$ | Calving flux | CCALVING |



| $F_{WFP}$ | Flood plains net upward water flux | CSRCFLOOD |
|:---:|:---:|:---:|
| (3b) | **HYD to OCE** | |
| $D_{is}$ | Coastal runoff | CRIVDIS |
| $F_{CGR}$ | Greenland Calving | CCALVGRE |
| $F_{CAN}$ | Antarctic Calving | CCALVANT |

Table 1: List of exchanged fields (see flow chart in Fig. 1)

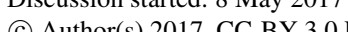


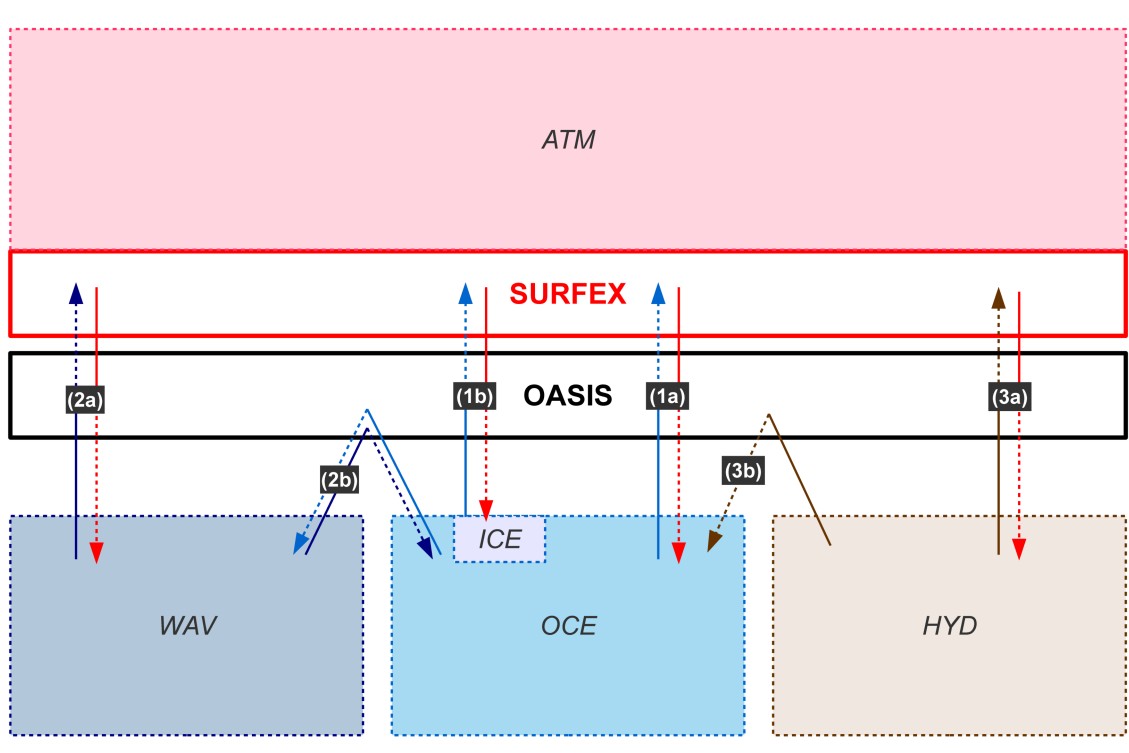

**Figure 1.** Flow chart of the SURFEX-OASIS coupling interface. An ocean model (OCE), possibly including a sea-ice model (ICE), a wave model (WAV) and an hydrological model (HYD) can exchange fields with the SURFEX interface (arrows 1a, 1b, 2a and 3a, resp.). Exchanges between OCE and WAV (and HYD resp.) are also possible through OASIS (arrows 2b and 3b resp.). When using an ATM component, SURFEX is embedded in the ATM executable.


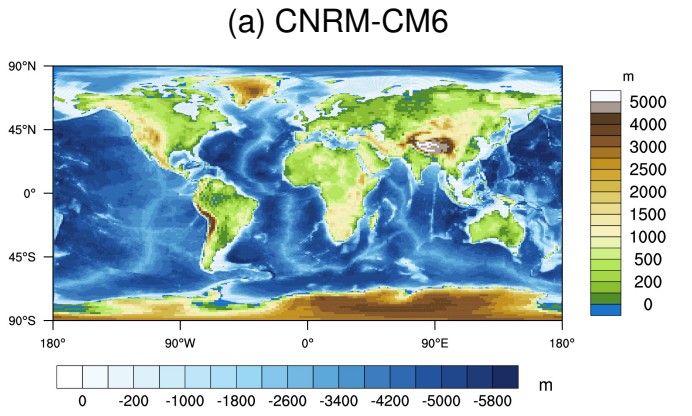

(a) CNRM-CM6

(b) CNRM-RCSM6

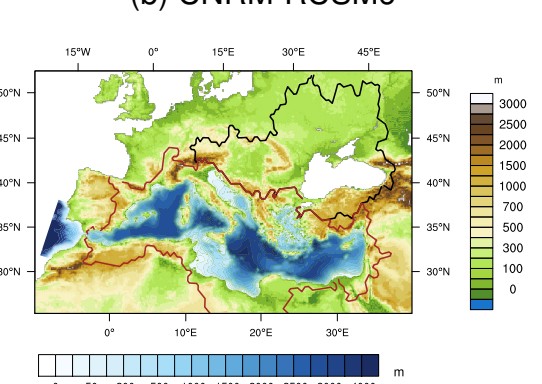

(c) AROME-NEMO WMED

(d) MESONH-MARS3D-WW3

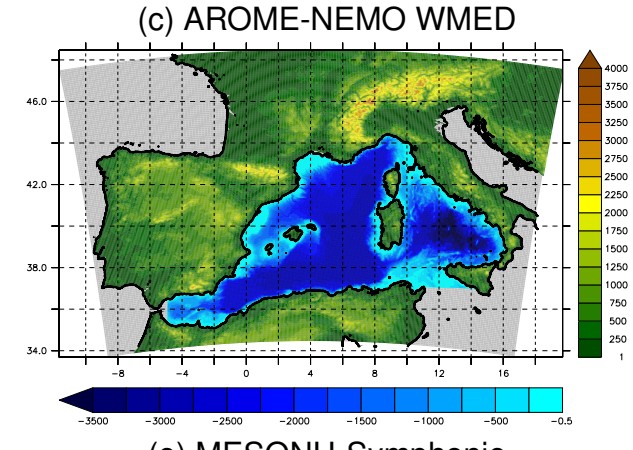

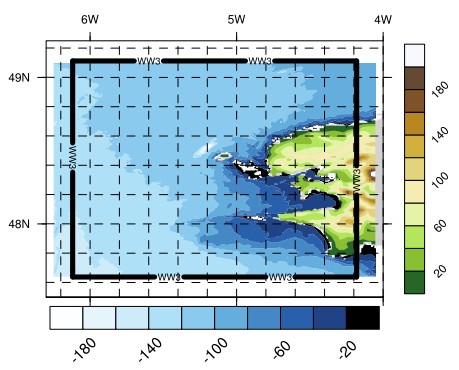

(e) MESONH-Symphonie

(f) MESONH-NEMO

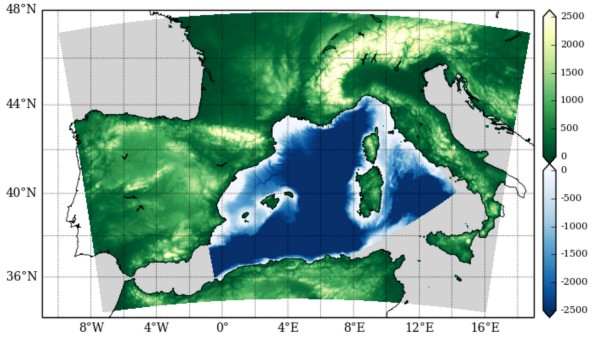

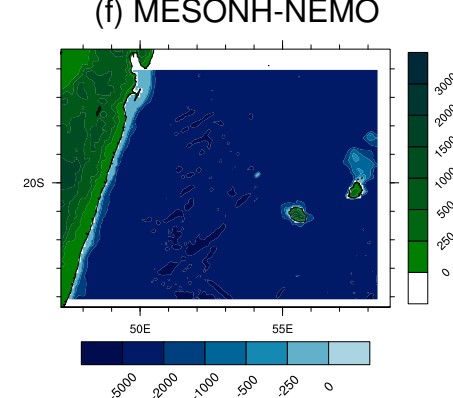



**Figure 2.** Domains of the limited-area coupled systems: topography (green colours) and bathymetry (blue colours) of the respective models **(a)** CNRM-CM6, **(b)** CNRM-RCSM6, **(c)** AROME-NEMO WMED, **(d)** MESONH-MARS3D-WaveWatchIII (Ouessant), **(e)** MESONH-Symphonie over the Western Mediterranean Sea, **(f)** MESONH-NEMO over the south-eastern Indian Ocean.

Discussion Paper | Discussion Paper | Discussion Paper | Discussion Paper

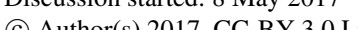



**Figure 3.** JJA mean land surface evaporation averaged over the period 1980-2009. On the zonal mean plot, the black line represents observationally derived data from Jung et al. (2009) averaged over 1982-2008, the red line the simulation.



**Figure 4.** Amplitude in °C of the 2009-2013 SST diurnal cycle for the Lion Buoy (x-axis) and the model simulation (y-axis), according to the season. Only values above 0.1° are kept. Mean values and standard deviations are given for each season, as well as the daily temporal correlation.



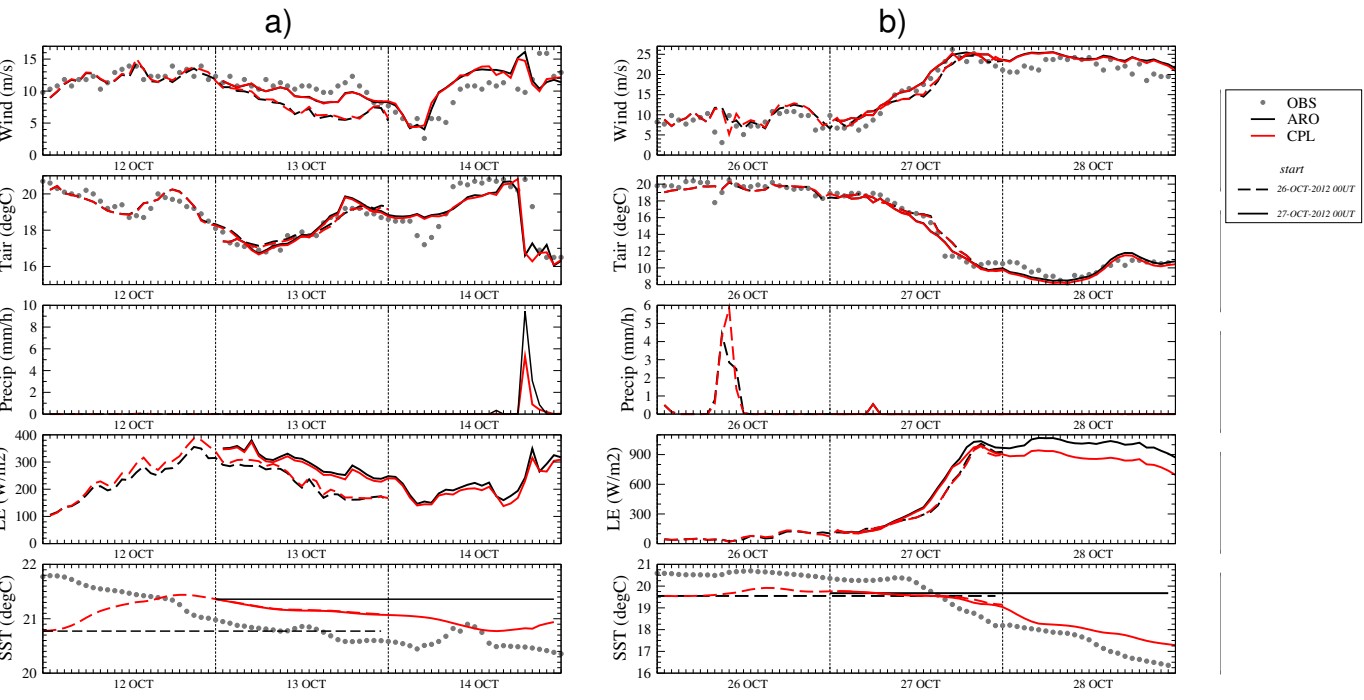

**Figure 5.** Time-series of wind (first row), temperature at first atmospheric level (second row), the surface precipitation rate (third row), latent heat flux (fourth row) and SST at LION buoy (fifth row) from the CPL (red line) and ARO forecasts (black line) during IOP13 (a) and IOP16b (b), compared to in-situ observations when available



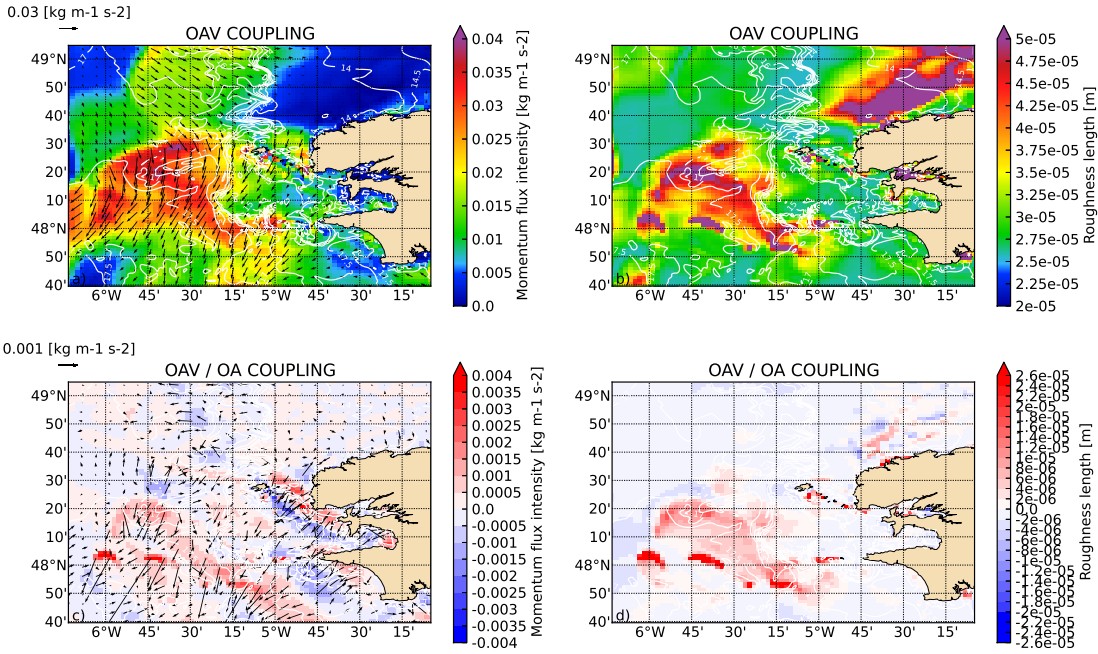

**Figure 6.** a) Wind stress module (color) and vector (arrows) (atmospheric sign convention), and SST (white contours with 0.5℃ interval) simulated by the fully ocean-atmosphere-waves coupling ; b) Roughness length (color) and SST (white contours with 0.5℃ interval) simulated by the fully ocean-atmosphere-waves coupling; c) Differences of the wind stress module (color) and vector (arrows) when simulated by the fully ocean-atmosphere-waves coupling versus the coupled atmospheric and oceanic dynamics alone ; d) Differences of the roughness length (color) and the SST (white contours with 0.5℃ interval) simulated by the fully ocean-atmosphere-waves coupling versus the coupled atmospheric and oceanic dynamics alone. All fields are shown at 9h, 2 hours after high tide.

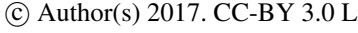


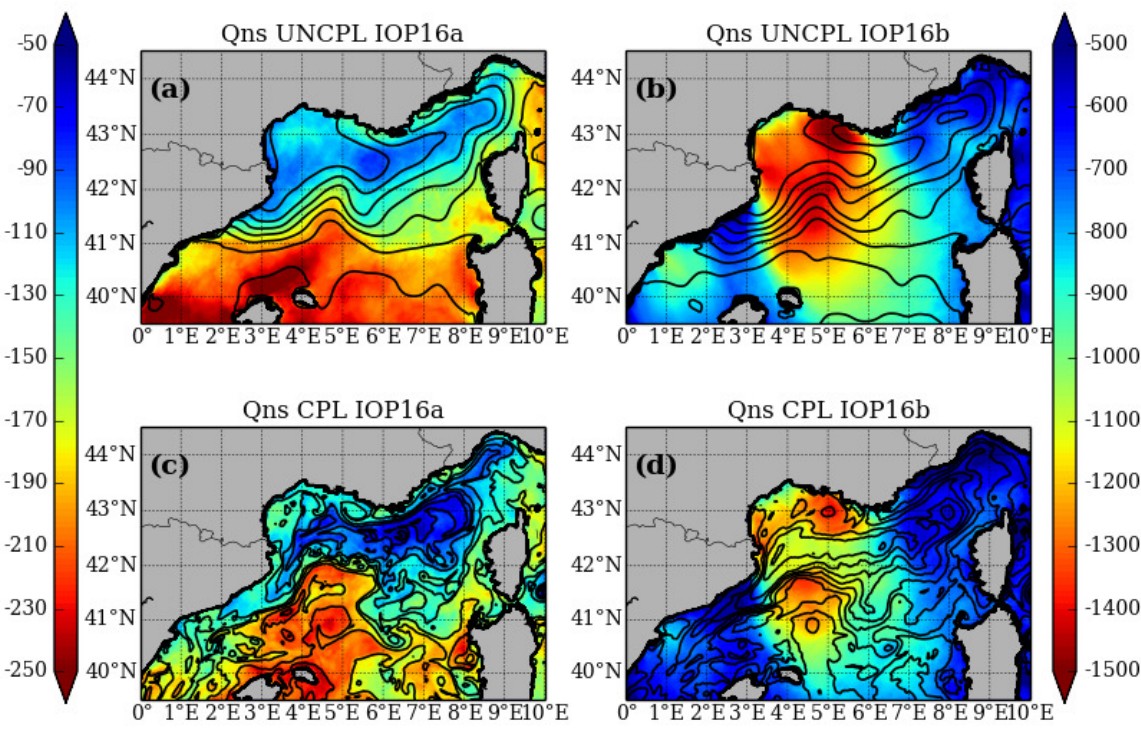

**Figure 7.** Spatial distribution of Non-solar heat flux (Qns, $W.m^{-2}$, see color-scale at left for (a) and (c) and at right for (b) and (d)) and SST in black contours (contour interval of 0.5℃) averaged over IOP16a ((a) and (c)) and IOP16b ((b) and (d)) for uncoupled simulation ((a) and (b)) and coupled simulation ((c) and (d)).



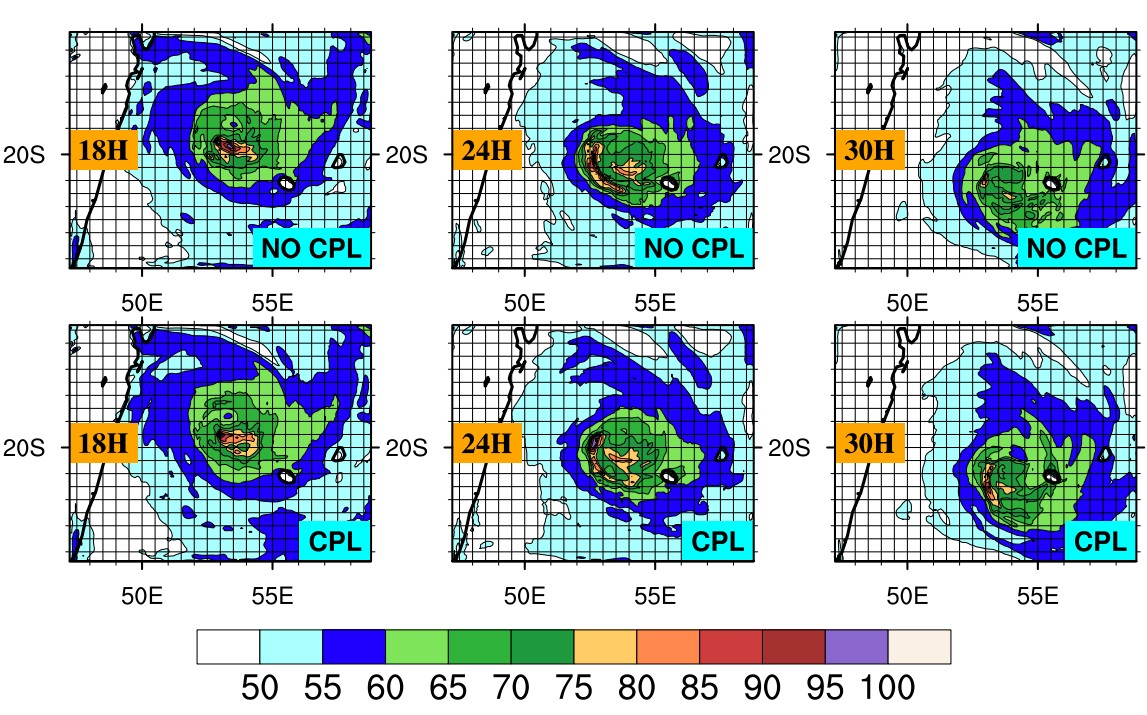

**Figure 8.** Total integrated water content in mm after 18h, 24h and 30h hours of simulation from left to rigth, respectively. Top panels show the results for the non coupled system (NOCPL i.e atmosphere only) and the bottom panels show th results for the coupled system (CPL).