# Peer review of "SURFEX v8.0 interface with OASIS3-MCT to couple atmosphere with hydrology, ocean, waves and sea-ice models, from coastal to global scales"

_Geoscientific Model Development, 2017_

## Referee Comment (RC1) · Anonymous Referee #1 · 13 Jun 2017

Comments on the paper " The seamless and multi-model coupling between atmosphere, land, hydrology, ocean, waves and sea-ice models based on SURFEX surface model using OASIS3-MCT"

General comments The paper describes the capability of the surface model SURFEX to couple a whole range of different model components. The possibility to switch between different components i.e. for the ocean model with their individual pros and cons is a great advantage in climate research. Several examples on the usage of SURFACE on different scales are described in the paper. The title is not so well chosen, the

word "seamless" does not appear in the paper. And its meaning is still unclear after reading the paper. Is it possible to use SURFEX in a coupled global model with two-way nesting refinements to have a seamless model in the spatial sense? Or is it meant in a temporal sense? If seamless is meant as the usage of SURFACE from global to high resolution models, the wording is not appropriate. The general focus of the paper is the description of the coupling interface and not to discuss the advantages of coupled modelling. But reading the paper gives a different impression. The examples of the different applications with SURFEX (Chap 3.) take a lot of space and they are only a loose listing of studies, where in all examples different variables are discussed. The frame is not so well defined as they all seems to use different SURFEX versions (Version 8, Version 7.2 and Version 7.3) and the differences are unclear. According to the introduction, the standard interface was introduced in Version 7.3. Please discuss the differences and the reason why there not using the same version. The conclusion fits well to the intention of the paper and it would have been better to focus more on the points in the conclusion as most of them are not discussed in the paper itself (i.e. computational costs of the coupling, grid interpolation, etc. ). As it is a more technical paper, it would have been better to discuss the technical aspects a bit clearer. Why is the atmospheric coupling omitted and which model component is calculating which fluxes and what problems could arise. This issue is superficially discussed in the conclusion, but it is very important for the coupling philosophy. And what about a two way nesting in the atmosphere with 3D coupling? This is not part of the SURFEX interface nor of the discussion (see seamless comment above). The approach of a general interface seems to be very tempting. What would be the problems to adapt the models for simulations i.e. over Africa? This is not clear to the user after reading the article. Every component seems to be available with certain license agreement, so how much more work is necessary? The last sentence of the conclusions discusses the question about the initialization of the coupled system. It would be better to discuss the advantages of the SURFEX community a bit earlier and maybe in more details. The discussion could also be centered around the interpolation of different grids and

the lateral boundary conditions in regional models. So the last sentence seems to be not really part of the rest of the paper. Please embed and write a nicer paragraph on this subject as it is very important when using SURFEX to have a community behind with a lot of support and possibilities for discussions and strategies for the future model development of SURFEX. There are a lot of citations in the article where the authors give credit to related work. The number and quality of the references are appropriate. The abstract give a good overview over the article, the discussion of the interface with its pros and cons could be added. The language is fluent and precise.

Specific comments In the introduction, page 3, line 84: this sentence is unclear. Why and when is the surface parametrization implemented in SURFEX not valid? Which 0D cases are available? For radiation studies? Please explain this in more detail and maybe not in the introduction. Page 5, line 122: "The fluxes are aggregated of the model grid-mesh". Are the aggregated fluxes passed directly to the corresponding model? What about a tile approach? First pass on the state variables for each tile to the corresponding model and then calculated the fluxes for each tile separately and average in the end. (See Barthélemy et al 2016). Page 5, line 140: What do the numbers refer to? Kind of obvious, but please use them first before you refer to them. Page 6, line 150: Where is this discussed again? This was not obvious in sections 3.2 or 3.3. It could be found in between the lines but please write a proper discussion about it. This is actually a very important subject for a coupling interface. How are the grids defined and what additional work is needed to adapt them to specific needs. Page 6, line 156: What is SYMPHONIE (mentioned later but not here). Page 15, line 441-442: This is an important issue and could be discussed also in more details and show up in the conclusions. It seems to be that the wave model so far is only coupled dynamically to ocean models and not to the atmosphere. Maybe it could be already mentioned in 2.3.2?

Technical corrections The size of the pdf is much too large. The figures, especially Fig 1 and some of Fig 2 seem to be enormous. Please reduce the size substantially,

there is no scientific gain to have a super high resolution topography figure with several layers in this pdf.

Barthélemy, A., Fichefet, T., Goosse, H., & Madec, G. (2016). A multi-column vertical mixing scheme to parameterize the heterogeneity of oceanic conditions under sea ice. Ocean Modelling, 104, 28-44.

---

## Referee Comment (RC2) · Anonymous Referee #2 · 21 Jun 2017

General comments:

The paper describes the coupling of the SUFREX surface model to a series of models using the OASIS3-MCT coupler different applications.

The title is, in my opinion, misleading. To me the word "seamless" means application of the same model to different time scales which is not at all what is described in the paper. I suggest to replace the word seamless with "versatile" which, again in my opinion, better cover the fact that the same surface model (SUFREX) can be used to

combine different atmospheric models with different ocean models, wave models and hydrological models.

Section 2 describes the principle of coupling with OASIS3-MCT and is the most useful part of the paper.

Section 3 describes the various coupled models implemented with the SURFEX OASIS3-MCT coupling. This part is a bit repetitive with a lot of model this version that. Maybe all the various models and versions could be presented in a table instead. In line with the comment about on the title of the paper I also find the "multi-model and multi-scale" title misleading. To me the concept of multi-model means the use of multiple different models to the same problem (e.g. seasonal forecasting) with each of the models add information to the solution of the problem. The various examples of performance of the various modelling system in section 3 needs justification. The examples don't need to be changed, but I do miss a justification why they were chosen for the various modelling systems.

The equations needs some work. An example: Eq 7 looks like an imported picture and very different from Eq 8. Another example is that Eq 6 looks very different from Eq 1 even though they are very similar. The quantities from the equations should be in italics in when they are explained in the text. So I believe a careful rework of the equations are in order.

The way the physical quantities with units are presented is very inconsistent throughout the manuscript. As an example on line 405 is written: "...goes from 10 m near the surface to 700m at the model top...". There are too many examples on these inconsistencies to list in a review, so I think that the authors needs to read the GMD guidelines (http://www.geoscientific-model-development.net/for_authors/manuscript_preparation.html) and adhere to those (e.g. 700m should be 700 m).

References: There is a lot of typo's and spelling errors in the references, so they could

really use a careful checking. Also the use of colours and/or underlining for links to DOI's, URL's or sometime authors is very inconsistent throughout.

In summary: I think that the manuscript has quite some merits and is worth publishing, but it is a bit let done by lack of care to details which is far from publication quality.

Specific comments:

Line 45-46: It might be worth mentioning that coupled NWP has been implemented operationally since 2013 at ECMWF in their ensemble system. It is described in Janssen et al ECMWF Tech Memo 712.

Line 59: Sequential versus concurrent coupling also affect the time stepping of the whole coupled modelling system. If accumulated fluxes are used then the flux user component cannot run until the producer component has accumulated the fluxes unless lagged fluxes are used. It might be worth to make this point.

Line 99+101: SURFEX versioning conventions are inconsistent across V7.3 or v8

Line 135: The subroutine names below are not in italics on my print.

Line 203: It might reflect conventions of different communities, but both zonal/meridional and x/y is used to present vector components.

Line 210: za should be z subscript a. Also keep in mind the general comments on equations.

Line 244: Hs should be H subscript s.

Line 247: There is no mentioning on sea-ice in the coupling between ocean and waves. Are the sea-ice wave interactions (an active research topic) not taken into account yet? Section 3.2: The normal convention is to use DJF, MAM, JJA, SON for seasons, but the authors have chosen JFM, AMJ, JJS, OND. Why?

Line 364: I believe that the ORCA grid family denotes tri-polar global grids.

[Figure]

Line 374: UT should be UTC.

Line 452: Why is the "Marshall and Schott 1999" underlined and in blue?

Line 473: Replace "no longer coupled" with "uncoupled".

Line 526: Why does global conservation make the model expensive?

Table 1: Charnock name is just a square. Meridian should be meridional.

Figure 5: The right hand text box only applies to b) since it explicitly states the 26/27 of October. It should be changed to e.g. "first/second".

Figure 6: Bottom panels: OAV/OA should be OAV – OA since this is actually what is plotted. The number on the lower legend are very close (special on the right hand plot). I also find that the white colour for the SST contours a poor choice making it very hard to see these.

---

## Referee Comment (RC3) · Anonymous Referee #3 · 24 Jun 2017

General comments:

The manuscript describes the new coupling interface integrated in the SURFEX surface model. This interface allows SURFEX to communicate through OASIS3-MCT coupler with other models with similar coupling interfaces. After the technical description of the interface, different applications and illustrations are presented. The main criticism concerning the manuscript in my opinion is the irregularity of the global layout and content quality: some sentences are too long or not clear, some acronyms are not detailed and some figures are incomplete or incorrectly referenced. The manuscript gives the feeling

that sections have been written by different authors and consequently lacks of consistency and homogeneity. A global and careful correction of the manuscript and the figures must be done by the authors to improve the manuscript reading, comprehension and consistency before publication. The main technical caveat is the lack of discussion about the computational performance of the new coupling interface (OASIS3-MCT in SURFEX) compared to the old one (OASIS3 in the atmospheric model) or to alternative coupling strategies such as the integration of all components in one executable. This could make the manuscript more useful for other modelling groups using different coupling strategies. Some models described in the manuscript allow to do grid-nesting (NEMO with AGRIF, MESO-NH for example). Is the SURFEX coupling interface compatible with such type of coupled model configurations? This possibility or limitation should be discussed in the text. A lot of different models are cited in the text, especially for the atmosphere and the hydrology. A supplementary table summarizing this list by category could facilitate the manuscript reading.

Specific comments:

Introduction: The introduction is not well organized: examples should be given just after describing the corresponding coupling strategy (i.e. add the COAWST model example just after the description of the "one executable" approach). A sketch or a table summarizing and comparing the 3 possible coupling strategies (one executable / multiple executable / integrated application) could help to better understand the strategy detailed in the manuscript.

Section 2: A separate paragraph for the description of OASIS3-MCT coupler is missing in the manuscript. The paragraph in introduction section from l.95 to l.102 could be moved in Section 2 and merged with l.128 to l.134 to create a new subsection describing OASIS3-MCT. A comparison of performances between the old implementation of OASIS3-MCT in ARPEGE-climate and the new version in SURFEX interface could give valuable information and improve the manuscript. A comparison of performances between OASIS3 and OASIS3-MCT version, which doesn't need to use dedicated processes contrary to OASIS3, could also give useful information for the modellers reading the manuscript. It is not clear if SURFEX can be run in coupled mode with an atmospheric model through OASIS3-MCT at different resolutions or not (l.159). If SURFEX need always to be compiled and integrated with the atmospheric model executable at the same resolution, this limitation must be clearly stated somewhere in the manuscript. Regarding the ICE model, (l. 184-l.195) it is not clear if it is coupled as an independent model to SURFEX or if the ICE model is already included in the OCEAN model and then is not coupled directly to SURFEX (such as LIM3 ice model in NEMO for example). This must be clearly stated in the text.

Section 3: A table summarizing the different coupled configurations based on SURFEX, their components and the corresponding versions could facilitate the Section 3 comprehension. The section 3.1 describing how the different components of the ESM are tested and assembled is very interesting but incomplete. A more detailed description of the validation process of the different components before coupling them would greatly improve the manuscript technical interest. For example, a paragraph describing NEMO-SURFEX evaluation before their coupling with the other components. The analysis of the diurnal cycle in CNRM-RCSM6 is interesting. Did you perform any analysis concerning the surface heat budget to understand why the diurnal cycle is overestimated in summer and underestimated in winter? Is it related to turbulent fluxes / radiative fluxes / etc. . . ? The MESONH-SYMPHONIE section is a bit redundant with the AROME-NEMO section because it uses the same dataset from Hymex IOP. Consequently, dataset description and the low-resolution simulations which is not presented in the manuscript can be removed to make this section easier to read. Concerning the Figure 7, a third column presenting the differences between CPL and UNCPL would facilitate the comparison discussed in the text. Concerning the MESONH-NEMO section, the illustration of the coupling effect on the tropical cyclone structure is not clear (and the units are missing). A better illustration would be to present the time evolution of the cyclone intensity or the water content averaged over the domain region.

Technical corrections:

Figure 2: remove "limited-area" from the legend as CNRM-CM6 is global and use the same color bar for all the panels.

Figure 3 is not well numbered.

Figure 6: Panel are not numbered and the white iso-contours are not visible in the lower panels.

Figure 8: Units are missing

l.1: Remove ÂńÂăseamlessÂăÂż from the title which is not adapted in this context

l.46: work done by Hewitt et al. 2016 at Met Office is relevant here

l.47: from the -> from a

l.48-50: Please explain the pro and cons about the fully embedded coupling strategy compared to the multi-executable one to better understand the choices of the different modelling groups.

l.52: communicate -> exchange regrid -> interpolate data into different grids

l.55-59: the sentence should be divided in 2 to improve manuscript readability.

l.76: Ric chi -> Ricchi

l.81: Is there any attempt to use Surfex coupled only to an ocean model (without atmospheric model)? If yes, it can be added here. If no (because of technical restrictions), it must be stated in the text.

l.84: could you give a practical example about Surfex parameterization limitation in stand-alone model to improve the manuscript ?

l. 104: use -> use cases

l. 140-143: are SURFEX "OFFLINE" and SURFEX stand-alone modes the same ? If

yes, please use the same terms everywhere in the manuscript. It is also not clear why SURFEX need to communicate with OASIS in this mode. This should be detailed in the manuscript.

l. 146: called "namcouple"

l.169: typo in the stress equation: Us-Ua-> Ua-Us

l. 210: za -> za

l. 247: regarding the Dis term in the hydrological model, how is it considered by the ocean model? Precipitation?

l. 310: please add the typical value of CMIP5 models SSH drift. Is it possible to determine if this drift is also present in observations or just related to unrealistic water cycle in the model?

References:

Hewitt HT, Roberts MJ, Hyder P, et al (2016) The impact of resolving the Rossby radius at mid-latitudes in the ocean: Results from a high-resolution version of the Met Office GC2 coupled model. Geosci Model Dev 9:3655–3670. doi: 10.5194/gmd-9-3655-2016

---

## Author Comment (AC1) · 28 Jul 2017

We thank the anonymous reviewers for their very constructive comments on our manuscript. The reviewers comments have greatly help in improving the quality of the manuscript. Below, we answer point by point (in italic) to the reviewer comments. Note that given the three reviewers comments that mainly stressed the lack of uniformity of the manuscript, we have largely reorganised and reworded the original text. As a consequence, it was not possible to highlight the manuscript changes. In the response to the comments, we have indicated the lines where the comment is addressed

in the new manuscript (added as a supplementary material). We also summarize all the changes made at the end of this document (section by section).

Reviewer's Comment: The paper describes the capability of the surface model SUR-FEX to couple a whole range of different model components. The possibility to switch between different components i.e. for the ocean model with their individual pros and cons is a great advantage in climate research. Several examples on the usage of SUR-FACE on different scales are described in the paper. The title is not so well chosen, the word "seamless" does not appear in the paper. And its meaning is still unclear after reading the paper. Is it possible to use SURFEX in a coupled global model with two-way nesting refinements to have a seamless model in the spatial sense? Or is it meant in a temporal sense? If seamless is meant as the usage of SURFACE from global to high resolution models, the wording is not appropriate.

Authors' Answer: We agree that the term seamless was misleading here. We have therefore changed the title to: "SURFEX v8.0 interface with OASIS3-MCT to couple atmosphere with hydrology, ocean, waves and sea-ice models, from coastal to global scales" to better reflect the asymmetry (see below) in the system between atmosphere and the other components and to focus on the generic aspect of the interface which makes it usable at very different scales. RC: The general focus of the paper is the description of the coupling interface and not to discuss the advantages of coupled modelling. But reading the paper gives a different impression. The examples of the different applications with SURFEX (Chap 3.) take a lot of space and they are only a loose listing of studies, where in all examples different variables are discussed.

AA: We agree that section 3 is a long and heterogeneous list both in form and content. We have decided to keep this section since we would like to illustrate the diversity of applications that can take advantage of the new interface. However, we have tried to shorten and make it more uniform for all systems in terms of presentation. This was easier than in the original manuscript since some of the systems have now lead to published scientific studies, that can be cited. We have also added a new table (2) on

page 29 summarizing the different systems and indicated the number of cores used by each component of the coupled systems. We have also added at the beginning of section 3, a discussion on the motivation for listing these applications, but also on the coupling cost. This new part is found on lines 322-350.

RC: The frame is not so well defined as they all seems to use different SURFEX versions (Version 8, Version 7.2 and Version 7.3) and the differences are unclear. According to the introduction, the standard interface was introduced in Version 7.3. Please discuss the differences and the reason why there not using the same version.

AA: The interface was introduced in version 8, and back-phased in former versions because some applications use atmospheric models that are linked to specific version of SURFEX. It was easier to back-phase the coupling interface than to change the SURFEX version, since SURFEX coupling with atmospheric models is hard-coded. The SURFEX version in which the coupling has been developed is clarified at the beginning of section 2.3 (old 2.2). See lines 183-189. The explanation on the reason why the systems use different SURFEX versions is clarified in the new part added at the beginning of section 3, lines 349-350. The versions are also clearly listed in the new table 2 (page 29).

RC: The conclusion fits well to the intention of the paper and it would have been better to focus more on the points in the conclusion as most of them are not discussed in the paper itself (i.e. computational costs of the coupling, grid interpolation, etc. ). As it is a more technical paper, it would have been better to discuss the technical aspects a bit clearer. Why is the atmospheric coupling omitted and which model component is calculating which fluxes and what problems could arise. This issue is superficially discussed in the conclusion, but it is very important for the coupling philosophy.

AA: We agree that some technical aspects were missing in the original manuscript. The computational costs are now discussed in section 3 (see above). We have also added more details on the coupling methods and interpolation. We have also reorganized

the end of introduction and section 2 to clarify the interface description. By doing so, we have created a new section 2.2 to discuss shortly the capacities of OASIS3-MCT. Most of the technical aspects that were not discussed are in fact tunable thanks to OASIS3-MCT namelist. We do not intend to describe exhaustively the OASIS3-MCT capacities, which are well described in the User Guide (Valcke et al, 2015), but this section gives the information on what is tunable via OASIS3-MCT. For the atmospheric coupling we have clarified this point in the introduction. The main point is that the coupling between SURFEX and ATM is done in a hard-coded way using a call to the SURFEX interface. Originally, SURFEX has been externalized from the atmospheric model so as to ease the development and sharing of the surface parameterisations in Meteo-France models, but the implicit coupling limited the possibility to use a coupler. The implementation of SURFEX within ATM is described in Masson et al. (2013) based on Best et al. (2004) structure and is clearly out of the scope of the study.

RC: And what about a two way nesting in the atmosphere with 3D coupling? This is not part of the SURFEX interface nor of the discussion (see seamless comment above). The approach of a general interface seems to be very tempting.

AA: The question of the two way nesting is interesting and was not addressed nor in the implementation nor in the manuscript. Several developments are notably needing to use the coupling interface with grid-nesting. We have added a discussion on this point in the conclusion as a perspective. See lines 359-352 on the question of the two-way nesting.

RC: What would be the problems to adapt the models for simulations i.e. over Africa? This is not clear to the user after reading the article. Every component seems to be available with certain license agreement, so how much more work is necessary?

AA: We have added a discussion on this point in the conclusion. The question of changing the region for limited area exemples is mainly a question of adapting each component to the region, there is no specific work to be done for the coupling in itself,

other than what is described at the end of section 2.3 (old 2.2) that has been also detailed. See lines 541-546 in the conclusion.

RC: The last sentence of the conclusions discusses the question about the initialization of the coupled system. It would be better to discuss the advantages of the SURFEX community a bit earlier and maybe in more details.The discussion could also be centered around the interpolation of different grids and the lateral boundary conditions in regional models. So the last sentence seems to be not really part of the rest of the paper. Please embed and write a nicer paragraph on this subject as it is very important when using SURFEX to have a community behind with a lot of support and possibilities for discussions and strategies for the future model development of SURFEX.

AA: The conclusion has been modified. A discussion on the advantages of the SUR-FEX community is now added in the conclusion, lines 524-532. The paragraph dealing with initial (and lateral boundary) conditions has been modified in the conclusion, lines 563-566.

RC: There are a lot of citations in the article where the authors give credit to related work. The number and quality of the references are appropriate. The abstract give a good overview over the article, the discussion of the interface with its pros and cons could be added. The language is fluent and precise.

Specific comments

RC: In the introduction, page 3, line 84: this sentence is unclear. Why and when is the surface parametrization implemented in SURFEX not valid? Which 0D cases are available? For radiation studies?

AA: We agree that the sentence was not clear. The introduction has been largely revised and this point has been clarified. One example of 0D case is the ocean flux bulk formulae (no level and made at each grid point). However, 0D case is not mentioned anymore in the introduction as it was not necessary for the objectives of the section.

RC: Please explain this in more detail and maybe not in the introduction. Page 5, line 122: "The fluxes are aggregated of the model grid-mesh". Are the aggregated fluxes passed directly to the corresponding model? What about a tile approach? First pass on the state variables for each tile to the corresponding model and then calculated the fluxes for each tile separately and average in the end. (See Barthélemy et al 2016).

AA: SURFEX use a tiling approach as explained in section 2.1, so that the fluxes are aggregated to the atmospheric grid mesh (better explained lines 130-134 and new Eq. 1) The fluxes sent to the ocean model are those calculated only over the ocean fraction of the grid cell. This is now explained at the end of section 2.4.1 (old 2.3.1), lines 238-244.

RC: Page 5, line 140: What do the numbers refer to? Kind of obvious, but please use them first before you refer to them.

AA: Done

RC: Page 6, line 150: Where is this discussed again? This was not obvious in sections 3.2 or 3.3. It could be found in between the lines but please write a proper discussion about it. This is actually a very important subject for a coupling interface. How are the grids defined and what additional work is needed to adapt them to specific needs.

AA: We agree that this was not discussed clearly. We have added a discussion in section 2.3 (old section 2.2), lines 176-182.

RC: Page 6, line 156: What is SYMPHONIE (mentioned later but not here).

AA: The model reference has been added here and removed in section 3.

RC: Page 15, line 441-442: This is an important issue and could be discussed also in more details and show up in the conclusions.

AA: This issue concerning the parameterization of the wave effect on the surface roughness, the momentum flux and thus on the atmospheric boundary layer is now discussed

in section 2.4.2 (lines 271-274). In the conclusion (lines 356-358), we reassert this point indicating that surface flux parameterizations are critical physical schemes in the coupled mode and that the interface between SURFEX and OASIS3-MCT offers the opportunity to more easily test new parameterization in multi-model and coupled mode.

RC: It seems to be that the wave model so far is only coupled dynamically to ocean models and not to the atmosphere. Maybe it could be already mentioned in 2.3.2?

AA: The wave model is coupled both with ocean and atmosphere models as described in section 2.3.2. The atmospheric roughness length is estimated at the air-sea interface by Equation 8. The Charnock parameter is directly calculated by the wave model for the M2W system and estimated from the wind in the other model configurations.

Technical corrections

RC: The size of the pdf is much too large. The figures, especially Fig 1 and some of Fig 2 seem to be enormous. Please reduce the size substantially, there is no scientific gain to have a super high resolution topography figure with several layers in this pdf.

AA: The size of Figure 1 has been decreased. Several plots of Figure 2 have been modified.

Barthélemy, A., Fichefet, T., Goosse, H., & Madec, G. (2016). A multi-column vertical mixing scheme to parameterize the heterogeneity of oceanic conditions under sea ice. Ocean Modelling, 104, 28-44.

References:

Best, M. J., Beljaars, A., Polcher, J., and Viterbo, P., 2004: A Proposed Structure for Coupling Tiled Surfaces with the Planetary Boundary Layer. J. Hydrometeorol., 5, 1271–1278.
* * *
Summary of all changes made in the manuscript (and rationale for the changes):

[Figure]

Title: changed

Abstract: few rewording, mainly unchanged.

Introduction: beginning unchanged until line 45, then mainly a reorganisation of existing text to provide examples along the different types of interpolation, plus clarification of pros and cons. From line 80-108 : rewording and adds on the implementation to better state the motivation, clarify the atmosphere/SURFEX coupling.

Section 2: section 2.1, only slightly modified (rewording)

Creation of new section 2.2 on OASIS3-MCT to better describe the technical implementation and highlight the capabilities of the implementation. This new section is partly based on existing paragraph from section 2.1 and 2.3.

Section 2.3: Changed to detail the necessary work to be done to set up a new coupled model and clarify the SURFEX version in which the developments have been made. Section 2.4: The header is only reworded

section 2.4.1: The beginning is unchanged. Mainly adds for the ICE coupling. Also detail the interaction between the tiling and the coupling.

section 2.4.2: the first part on the ATM-WAV coupling has been reworded to better introduce the motivation and to clarify the methods. Second part on OCE-WAV coupling only reworded.

section 2.4.3 : only slight rewording.

Section 3: creation of new paragraphs before section 3.1 to clarify the motivation for the examples described in this section and discuss the cost of coupling. This comes with new table 2.

Section 3.4 and 3.6 have been switched to provide examples from the largest scale to the smaller one (figures 6 and 8 have been switched accordingly).

Sub-sections of section 3 have all been shortened and reworded for uniformization.

Conclusion: The conclusion has been largely revised to better discuss the pros and cons of the implementation. Some new perspectives have been added (new coupling fields, two-way nesting).

Authors contribution: unchanged

References: checked

Figures

Figure 1: unchanged

Figure 2: layout changed for uniformisation (and figures d and f switched)

Figure 3 unchanged

Figure 4: unchanged

Figure 5: simplified

Figure 6: add differences CPL-UNCPL (e and f)

Figure 7: remove one lag

Figure 8: improved following reviewer suggestions, captions reworded.

Tables

Table 1: rewording ("remove x and y components of")

New table 2

Please also note the supplement to this comment:
https://www.geosci-model-dev-discuss.net/gmd-2017-91/gmd-2017-91-AC1-supplement.pdf

**Supplement:**

[revised manuscript text omitted]

---

## Author Comment (AC2) · 28 Jul 2017

We thank the anonymous reviewers for their very constructive comments on our manuscript. The reviewers comments have greatly help in improving the quality of the manuscript. Below, we answer point by point (RC) to the reviewer comments. Note that given the three reviewers comments that mainly stressed the lack of uniformity of the manuscript, we have largely reorganised and reworded the original text. As a consequence, it was not possible to highlight the manuscript changes. In the response to the comments, we have indicated the lines where the comment is addressed in the new

manuscript (added as a supplementary material). We also summarize all the changes made at the end of this document (section by section).

Reviewer's Comment: The paper describes the coupling of the SUFREX surface model to a series of models using the OASIS3-MCT coupler different applications. The title is, in my opinion, misleading. To me the word "seamless" means application of the same model to different time scales which is not at all what is described in the paper. I suggest to replace the word seamless with "versatile" which, again in my opinion, better cover the fact that the same surface model (SUFREX) can be used to combine different atmospheric models with different ocean models, wave models and hydrological models.

Authors' answer: We agree that the title was not well chosen. We have decided to change it to: "SURFEX v8.0 interface with OASIS3-MCT to couple atmosphere with hydrology, ocean, waves and sea-ice models, from coastal to global scales" to better reflect the asymmetry in the system between atmosphere and the other components and to focus on the generic aspect of the interface which makes it usable at very different scales.

RC: Section 2 describes the principle of coupling with OASIS3-MCT and is the most useful part of the paper.

RC: Section 3 describes the various coupled models implemented with the SURFEX OASIS3-MCT coupling. This part is a bit repetitive with a lot of model this version that. Maybe all the various models and versions could be presented in a table instead. In line with the comment about on the title of the paper I also find the "multi-model and multi-scale" title misleading. To me the concept of multi-model means the use of multiple different models to the same problem (e.g. seasonal forecasting) with each of the models add information to the solution of the problem. The various examples of performance of the various modelling system in section 3 needs justification. The examples don't need to be changed, but I do miss a justification why they were chosen

for the various modelling systems.

AA: We agree that section 3 is a long and heterogeneous list both in form and content. We have decided to keep this section since we would like to illustrate the diversity of applications that can take advantage of the new interface. However, we have tried to shorten and make it more uniform for all systems in terms of presentation. This was easier than in the original manuscript since some of the systems have now lead to published scientific studies, that can be cited. We have also added a new table 2 on page 29 summarizing the different systems and indicated the number of cores used by each component of the coupled systems. We have also added at the beginning of section 3, a discussion on the motivation for listing these applications, but also on the coupling cost. This new part is found on page 11, lines 322-379.

RC: The equations needs some work. An example: Eq 7 looks like an imported picture and very different from Eq 8. Another example is that Eq 6 looks very different from Eq 1 even though they are very similar. The quantities from the equations should be in italics in when they are explained in the text. So I believe a careful rework of the equations are in order.

AA: We agree that there were still some formal problems with the equations, we have worked on this before the final submission of the revised paper.

RC: The way the physical quantities with units are presented is very inconsistent throughout the manuscript. As an example on line 405 is written: ". . .goes from 10 m near the surface to 700m at the model top. . .". There are too many examples on these inconsistencies to list in a review, so I think that the authors needs to read the GMD guidelines (http://www.geoscientific-model-development.net/for_authors/manuscript_preparation.html) and adhere to those (e.g. 700m should be 700 m).

AA: For the revised version, we have checked the consistency of how the units are written and now it follows the GMD guidelines.

RC: References: There is a lot of typo's and spelling errors in the references, so they could really use a careful checking. Also the use of colours and/or underlining for links to DOI's, URL's or sometime authors is very inconsistent throughout.

AA: The list of references has been carefully revised.

RC: In summary: I think that the manuscript has quite some merits and is worth publishing, but it is a bit let done by lack of care to details which is far from publication quality.

Specific comments:

RC: Line 45-46: It might be worth mentioning that coupled NWP has been implemented operationally since 2013 at ECMWF in their ensemble system. It is described in Janssen et al ECMWF Tech Memo 712.

AA: We have added the reference Janssen et al. 2013.

RC: Line 59: Sequential versus concurrent coupling also affect the time stepping of the whole coupled modelling system. If accumulated fluxes are used then the flux user component cannot run until the producer component has accumulated the fluxes unless lagged fluxes are used. It might be worth to make this point.

AA: We have added the sentence:"For example, if the components are sequentially coupled (i.e. one component cannot do any work while the other is running to produce its coupling field and vice-versa), running concurrently on different sets of resources will lead to some waste of resources." to underline the impact of sequential vs concurrent coupling.

RC: Line 99+101: SURFEX versioning conventions are inconsistent across V7.3 or v8

AA: v8 has been replaced by v8.0.

RC: Line 135: The subroutine names below are not in italics on my print.

AA: This is corrected.

RC: Line 203: It might reflect conventions of different communities, zonal/meridional and x/y is used to present vector components. but both

AA: x/y have been removed.

RC: Line 210: za should be z subscript a. Also keep in mind the general comments on equations.

AA: Changed

RC: Line 244: Hs should be H subscript s.

AA: Done

RC: Line 247: There is no mentioning on sea-ice in the coupling between ocean and waves. Are the sea-ice wave interactions (an active research topic) not taken into account yet?

AA: This is not taken into account yet. We now mention in the conclusion this missing point that would require further investigations.

RC: Section 3.2: The normal convention is to use DJF, MAM, JJA, SON for seasons, but the authors have chosen JFM, AMJ, JJS, OND. Why?

AA: These seasons are the usual conventions for the ocean diagnostics.

RC: Line 364: I believe that the ORCA grid family denotes tri-polar global grids.

AA: Yes, but only a regional (irregular rectangular) fraction of the global grid is used. This mention of the ORCA grid was deleted in section 3.3.

RC: Line 374: UT should be UTC.

AA: Done

RC: Line 452: Why is the "Marshall and Schott 1999" underlined and in blue?

AA: There was a link in the electronic version, it has been removed.

RC: Line 473: Replace "no longer coupled" with "uncoupled".

AA: Done

RC: Line 526: Why does global conservation make the model expensive?

AA: This remark has been removed since there is a new option in OASIS3-MCT that enables to speed up such calculations. The main problem for global conservations calculations is that it necessitates to recombine information from all processes and this can hinder the parallelism efficacy.

RC: Table 1: Charnock name is just a square. Meridian should be meridional.

AA: Done

RC: Figure 5: The right hand text box only applies to b) since it explicitly states the 26/27 of October. It should be changed to e.g. "first/second".

AA: The terms "first day (D)" and "second day (D+1)" are now put in the text box and the corresponding dates for (a) and (b) are indicated in the legend.

RC: Figure 6: Bottom panels: OAV/OA should be OAV – OA since this is actually what is plotted. The number on the lower legend are very close (special on the right hand plot). I also find that the white colour for the SST contours a poor choice making it very hard to see these.

AA: The Figure 8 (old Fig. 6) has been modified following your comment and guidelines.
* * *
Summary of all changes made in the manuscript (and rationale for the changes):

Title: changed

[Figure]

Abstract: few rewording, mainly unchanged.

Introduction: beginning unchanged until line 45, then mainly a reorganisation of existing text to provide examples along the different types of interpolation, plus clarification of pros and cons. From line 80-108 : rewording and adds on the implementation to better state the motivation, clarify the atmosphere/SURFEX coupling.

Section 2: section 2.1, only slightly modified (rewording) Creation of new section 2.2 on OASIS3-MCT to better describe the technical implementation and highlight the capabilities of the implementation. This new section is partly based on existing paragraph from section 2.1 and 2.3.

Section 2.3: Changed to detail the necessary work to be done to set up a new coupled model and clarify the SURFEX version in which the developments have been made.

Section 2.4: The header is only reworded

section 2.4.1: The beginning is unchanged. Mainly adds for the ICE coupling. Also detail the interaction between the tiling and the coupling.

section 2.4.2: the first part on the ATM-WAV coupling has been reworded to better introduce the motivation and to clarify the methods. Second part on OCE-WAV coupling only reworded.

section 2.4.3 : only slight rewording.

Section 3: creation of new paragraphs before section 3.1 to clarify the motivation for the examples described in this section and discuss the cost of coupling. This comes with new table 2.

Section 3.4 and 3.6 have been switched to provide examples from the largest scale to the smaller one (figures 6 and 8 have been switched accordingly).

Sub-sections of section 3 have all been shortened and reworded for uniformization.

Conclusion: The conclusion has been largely revised to better discuss the pros and cons of the implementation. Some new perspectives have been added (new coupling fields, two-way nesting).

Authors contribution: unchanged

References: checked

Figures

Figure 1: unchanged

Figure 2: layout changed for uniformisation (and figures d and f switched)

Figure 3 unchanged

Figure 4: unchanged

Figure 5: simplified

Figure 6: add differences CPL-UNCPL (e and f)

Figure 7: remove one lag

Figure 8: improved following reviewer suggestions, captions reworded.

Tables

Table 1: rewording ("remove x and y components of")

New table 2

Please also note the supplement to this comment:
https://www.geosci-model-dev-discuss.net/gmd-2017-91/gmd-2017-91-AC2-supplement.pdf
* * *
[Figure]

2017.

---

## Author Comment (AC3) · 28 Jul 2017

We thank the anonymous reviewers for their very constructive comments on our manuscript. The reviewers comments have greatly help in improving the quality of the manuscript. Below, we answer point by point (RC) to the reviewer comments. Note that given the three reviewers comments that mainly stressed the lack of uniformity of the manuscript, we have largely reorganised and reworded the original text. As a consequence, it was not possible to highlight the manuscript changes. In the response to the comments, we have indicated the lines where the comment is addressed in the new

manuscript (added as a supplementary material). We also summarize all the changes made at the end of this document (section by section).

Reviewer's Comment: The manuscript describes the new coupling interface integrated in the SURFEX surface model. This interface allows SURFEX to communicate through OASIS3-MCT coupler with other models with similar coupling interfaces. After the technical description of the interface, different applications and illustrations are presented. The main criticism concerning the manuscript in my opinion is the irregularity of the global layout and content quality: some sentences are too long or not clear, some acronyms are not detailed and some figures are incomplete or incorrectly referenced. The manuscript gives the feeling that sections have been written by different authors and consequently lacks of consistency and homogeneity. A global and careful correction of the manuscript and the figures must be done by the authors to improve the manuscript reading, comprehension and consistency before publication. The main technical caveat is the lack of discussion about the computational performance of the new coupling interface (OASIS3-MCT in SURFEX) compared to the old one (OASIS3 in the atmospheric model) or to alternative coupling strategies such as the integration of all components in one executable.

Authors' answer: We agree that the question of the computational performance of the coupling was not really discussed. We have tried to rationalized section 3, to shorten the examples and we have added a discussion at the beginning of the section to highlight the objectives of this section. We have also added a discussion on the computational cost of the coupling in these examples and a table (table 2, page 29) giving the relative cost of each model component in the coupled system. We agree with the reviewer that it would have been interesting to provide the cost of using OASIS3 instead of OASIS3-MCT, however, this has not been possible in the context of SURFEX. This would have only been applicable to the climate applications which already existed before creating the new interface but this change came along with a change of all components versions and the interface with the old coupler was removed from the last

versions of the model. Additionally, the intermediate versions in which it would have been possible to make an evaluation, have not been ported to the new super-computer and are now unusable.

Separate performance tests have shown that OASIS3-MCT is much more efficient than the previous sequential OASIS3 version. Figure 4 in Craig et al. (2017) shows that the time for a back-and-forth coupling exchange between a T799 grid (i.e. a global atmospheric gaussian reduced grid with 843,490 grid points) and an ORCA025 grid (i.e a tripolar grid with $1442 \times 1021$ grid points) is about an order of magnitude smaller in OASIS3-MCT for a large range of core counts. The coupling cost has not been very well documented in all applications because it is not straightforward and mainly because none of the group did faced with performance problems. As theses coupling models were implemented for research purposes, the first versions of the models were not carefully checked in terms of cost. Such a work is underway (but with a different quickness in the different teams) and OASIS3-MCT comes with a tool (lucia) to estimate the coupling costs. This work has been done for the CMIP6 climate model and the cost of the coupling with OASIS3-MCT is very limited if the components are adapted to run with a similar speed so that none of the component wait before coupling exchanges.

The reason why we have chosen the coupler strategy is clearly for flexibility and we know that it is of major importance for SURFEX that is used in many different models with various components developed in different teams. Given that the coupling cost is not important compared to the individual model cost, the comparison with an hard-coded coupling is out of the scope of our model development.

A discussion on the motivation for the applications and on the coupling cost have been added at the beginning of section 3 (lines 322-379).

RC: This could make the manuscript more useful for other modelling groups using different coupling strategies. Some models described in the manuscript allow to do

grid-nesting (NEMO with AGRIF, MESO-NH for example). Is the SURFEX coupling interface compatible with such type of coupled model configurations? This possibility or limitation should be discussed in the text.

AA: Several developments are needed before using the coupling interface with grid-nesting, notably to manage with OASIS and in the four steps indicated in section 2.3 (initialization, definition, exchanges, finalization) the use of several domains by one executable. A discussion about this issue has been added in the conclusion. See lines 559-562 on the question of the two-way nesting.

RC: A lot of different models are cited in the text, especially for the atmosphere and the hydrology. A supplementary table summarizing this list by category could facilitate the manuscript reading.

AA: The list of models has been added in the new table 2 (page 29).

Specific comments:

RC: Introduction: The introduction is not well organized: examples should be given just after describing the corresponding coupling strategy (i.e. add the COAWST model example just after the description of the "one executable" approach). A sketch or a table summarizing and comparing the 3 possible coupling strategies (one executable / multiple executable / integrated application) could help to better understand the strategy detailed in the manuscript.

AA: We have reorganised the introduction as suggested by the reviewer.

RC: Section 2: A separate paragraph for the description of OASIS3-MCT coupler is missing in the manuscript. The paragraph in introduction section from l.95 to l.102 could be moved in Section 2 and merged with l.128 to l.134 to create a new subsection describing OASIS3-MCT.

AA: We have created a new section 2.2 dedicated to OASIS3-MCT as suggested by the reviewer. This new section aims at providing the information of what OASIS3-MCT

handles but a complete description can be founded in OASIS3-MCT documentation (Valcke et al., 2015). Accordingly, we have fully reorganised section 2.

RC: A comparison of performances between the old implementation of OASIS3-MCT in ARPEGE-climate and the new version in SURFEX interface could give valuable information and improve the manuscript. A comparison of performances between OASIS3 and OASIS3-MCT version, which doesn't need to use dedicated processes contrary to OASIS3, could also give useful information for the modellers reading the manuscript.

AA: We have discussed this point above.

RC: It is not clear if SURFEX can be run in coupled mode with an atmospheric model through OASIS3-MCT at different resolutions or not (l.159). If SURFEX need always to be compiled and integrated with the atmospheric model executable at the same resolution, this limitation must be clearly stated somewhere in the manuscript.

AA: When coupled to an atmospheric models, SURFEX operates at the atmospheric model resolution and the coupling is hard-coded. This point is now clearly stated at the end of introduction (lines 84-85).

RC: Regarding the ICE model, (l. 184-l.195) it is not clear if it is coupled as an independent model to SURFEX or if the ICE model is already included in the OCEAN model and then is not coupled directly to SURFEX (such as LIM3 ice model in NEMO for example). This must be clearly stated in the text.

AA: The coupling interface does not make any hypothesis on this point. SURFEX can receive the ICE fields from an independent ICE model, this is managed through the OASIS namelist. This is now stated at the end of section 2.4.1 (old 2.3.1), lines 235-237.

RC: Section 3: A table summarizing the different coupled configurations based on SURFEX, their components and the corresponding versions could facilitate the Section 3 comprehension.

AA: A new table 2 (page 29) has been created as suggested by the reviewer.

RC: The section 3.1 describing how the different components of the ESM are tested and assembled is very interesting but incomplete. A more detailed description of the validation process of the different components before coupling them would greatly improve the manuscript technical interest. For example, a paragraph describing NEMO-SURFEX evaluation before their coupling with the other components.

AA: We agree that this would have been valuable but given the remarks of all the reviewers, we have chosen to shorten and uniformize the examples given in section 3, rather than to add more details.

RC: The analysis of the diurnal cycle in CNRM-RCSM6 is interesting. Did you perform any analysis concerning the surface heat budget to understand why the diurnal cycle is overestimated in summer and underestimated in winter? Is it related to turbulent fluxes / radiative fluxes / etc...?

AA: First, we apologize for the mistake on the season: the amplitude of the diurnal cycle is underestimated in OND which is autumn and not winter. Further analyzes on the reasons for the overestimation [underestimation] in summer [autumn] are planned. Our preliminary investigations indicate that the shortwave flux is overestimated by the model in summer. For what concerns the autumn, the wind representation for example should be studied in more detail.

RC: The MESONH-SYMPHONIE section is a bit redundant with the AROME-NEMO section because it uses the same dataset from Hymex IOP. Consequently, dataset description and the low-resolution simulations which is not presented in the manuscript can be removed to make this section easier to read.

AA: This section has been reorganized and reduced and motivations have been added et the beginning of section 3 (namely this validates the use of a different ocean model).

RC: Concerning the Figure 7, a third column presenting the differences between CPL

and UNCPL would facilitate the comparison discussed in the text.

AA: Done on Figure 6 (old Fig. 7)

RC: Concerning the MESONH-NEMO section, the illustration of the coupling effect on the tropical cyclone structure is not clear (and the units are missing). A better illustration would be to present the time evolution of the cyclone intensity or the water content averaged over the domain region.

AA: The evolution of the domain-averaged integrated water content is shown in Figure 1 (below, in this document) for the MESONH-NEMO model over the Indian Ocean. It shows that in the CPL run, there is systematically slightly more water than in the NOCPL run, but it does not show the impact on the structure of the system, and especially on the position and intensity of the eyewall represented by the region of maximum values of total water, which is what we want to illustrate. So, to render Figure 7 (old Fig. 8) more readable and in order also to enlarge the size of each plot, we have decreased the number of panels shown from 6 to 4. Only the 18 h and 30 h forecast ranges have been retained. The units are also shown on Figure 7.

Figure A: Domain-averaged total water content (mm) for the NOCPL run (in red) and for the CPL run (in blue).

Technical corrections:

RC: Figure 2: remove "limited-area" from the legend as CNRM-CM6 is global and use the same color bar for all the panels.

AA: Done

RC: Figure 3 is not well numbered.

AA: Corrected

RC: Figure 6: Panel are not numbered and the white iso-contours are not visible in the lower panels.

AA: Figure 8 (old Fig. 6) has been modified following your comment and guidelines.

RC: Figure 8: Units are missing

AA: Done in Figure 7 (old Fig. 8)

RC: l.1: Remove ń ăseamless ă Åij from the title which is not adapted in this context

AA: The title has been reworded to: "SURFEX v8.0 interface with OASIS3-MCT to couple atmosphere with hydrology, ocean, waves and sea-ice models, from coastal to global scales".

RC: l.46: work done by Hewitt et al. 2016 at Met Office is relevant here

AA: We have added the reference.

RC: l.47: from the -> from a

AA: Done

RC: l.48-50: Please explain the pro and cons about the fully embedded coupling strategy compared to the multi-executable one to better understand the choices of the different modelling groups.

AA: The pros and cons of the strategy are now more detailed in the introduction (lines 57-60, 63-65, 75-77) and reasserted in the conclusion (531-533).

RC: l.52: communicate -> exchange regrid -> interpolate data into different grids

AA: Done

RC: l.55-59: the sentence should be divided in 2 to improve manuscript readability.

AA: We split the sentence. It now reads: "In the first category, coupling is achieved via component-level interfaces within one integrated application, e.g. the Earth System Modeling Framework (ESMF, http://www.earthsystemmodeling.org, Collins et al., 2005, Theurich et al, 2016) and requires users to split components into initialise, run, and

finalise parts. This approach limits the places where data exchanges can happen but offers opportunities for performance optimization as components can be easily run in different layouts on available resources."

RC: l.76: Ric chi -> Ricchi

AA: Done

RC: l.81: Is there any attempt to use Surfex coupled only to an ocean model (without atmospheric model)? If yes, it can be added here. If no (because of technical restrictions), it must be stated in the text.

AA: SURFEX stand-alone can be coupled to an ocean model. This is now clearly stated at the end of the introduction (lines 83-84) and referenced in the conclusion (lines 551-553)

RC: l.84: could you give a practical example about Surfex parameterization limitation in stand-alone model to improve the manuscript?

AA: This has been added to the introduction (lines 91-95).

RC: l. 104: use -> use cases

AA: Done

RC: l. 140-143: are SURFEX "OFFLINE" and SURFEX stand-alone modes the same? If yes, please use the same terms everywhere in the manuscript. It is also not clear why SURFEX need to communicate with OASIS in this mode. This should be detailed in the manuscript.

AA: We agree that this point was not clear in the manuscript. We have removed the OFFLINE word and used stand-alone everywhere instead. Additionally, the introduction has been detailed to better explain the way SURFEX is interfaced with the atmospheric model (hard-coded interface, lines 84-85).

RC: l. 146: called "namcouple"

AA: Done

RC: l.169: typo in the stress equation: Us-Ua-> Ua-Us

AA: It is not a typo as SURFEX uses a different convention than ocean models in general. This field is multiplied by -1 in the coupling step (and this is specified in the "namcouple" file).

RC: l. 210: za -> za

AA: Done

RC: l. 247: regarding the Dis term in the hydrological model, how is it considered by the ocean model? Precipitation?

AA: The ocean model takes into account a river discharge term in the coupling interface.

RC: l. 310: please add the typical value of CMIP5 models SSH drift. Is it possible to determine if this drift is also present in observations or just related to unrealistic water cycle in the model?

AA: This point has been removed from the manuscript because it was not properly done in the simulations presented here. The main reason is that the coupled model has not been run long enough in the examples provided to reach an equilibrium in term of water budget. Such an effort has been done since the submission of the manuscript and after $\sim$300 years of simulation the ssh drift is less than 10-5 mm per century. Even if this is an interesting point to discuss, it is quite complex (given the anthropogenic impact) and would necessitate a rather long discussion which was not the focus of the paper.

References:

[Figure]

Craig, A., S. Valcke, L. Coquart: Development and performance of a new version of the OASIS coupler, OASIS3-MCT_3.0. Geosci. Model Dev. Discuss., doi:10.5194/gmd-2017-64.
* * *
Summary of all changes made in the manuscript (and rationale for the changes):

Title: changed

Abstract: few rewording, mainly unchanged.

Introduction: beginning unchanged until line 45, then mainly a reorganisation of existing text to provide examples along the different types of interpolation, plus clarification of pros and cons. From line 80-108 : rewording and adds on the implementation to better state the motivation, clarify the atmosphere/SURFEX coupling.

Section 2: section 2.1, only slightly modified (rewording) Creation of new section 2.2 on OASIS3-MCT to better describe the technical implementation and highlight the capabilities of the implementation. This new section is partly based on existing paragraph from section 2.1 and 2.3.

Section 2.3: Changed to detail the necessary work to be done to set up a new coupled model and clarify the SURFEX version in which the developments have been made.

Section 2.4: The header is only reworded

section 2.4.1: The beginning is unchanged. Mainly adds for the ICE coupling. Also detail the interaction between the tiling and the coupling.

section 2.4.2: the first part on the ATM-WAV coupling has been reworded to better introduce the motivation and to clarify the methods. Second part on OCE-WAV coupling only reworded.

section 2.4.3 : only slight rewording.

**GMDD**

Section 3: creation of new paragraphs before section 3.1 to clarify the motivation for the examples described in this section and discuss the cost of coupling. This comes with new table 2.

Section 3.4 and 3.6 have been switched to provide examples from the largest scale to the smaller one (figures 6 and 8 have been switched accordingly).

Sub-sections of section 3 have all been shortened and reworded for uniformization.

Conclusion: The conclusion has been largely revised to better discuss the pros and cons of the implementation. Some new perspectives have been added (new coupling fields, two-way nesting).

Authors contribution: unchanged

References: checked

Figures

Figure 1: unchanged

Figure 2: layout changed for uniformisation (and figures d and f switched)

Figure 3 unchanged

Figure 4: unchanged

Figure 5: simplified

Figure 6: add differences CPL-UNCPL (e and f)

Figure 7: remove one lag

Figure 8: improved following reviewer suggestions, captions reworded.

Tables

Table 1: rewording ("remove x and y components of")

[Figure]

New table 2

Please also note the supplement to this comment:
https://www.geosci-model-dev-discuss.net/gmd-2017-91/gmd-2017-91-AC3-
supplement.pdf

—————————————————

Interactive
comment

[Figure]

**Fig. 1.**

[Figure]

---

## Author Response (AR2)

Comments of referee #1:

Comments on the paper " **SURFEX v8.0 interface with OASIS3-MCT to couple atmosphere with hydrology, ocean, waves and sea-ice models, from coastal to global scales**"

**Reviewer's Comment:**

General comments

The paper has improved a lot in the review process. Only some minor issues turned up.

**RC:** Page 2, lines 104ff and page 5, lines 136ff have similar sentences. Please rewrite this, so it is only mentioned once that oasis3 was already used. In the paragraph on page5, line 138, please change to the new OASIS citation Craig 2017 as is was done earlier in the text.

*AA: Done*

**RC:** Page 3 or 4: It would be nice to mention here that SURFEX is freely available.

*AA: Done ln 82.*

**RC:** Page 7, line 194, this sentence took me a while to understand. It reads like Nemo, Symphonie and MARS3D are independent models, so it might be better to split in two sentences, and explain the connection between NEMO and the models a bit better.

*AA: These ocean models are indeed independent. The sentence has been slightly rephrased "This was only verified for OCE, as **either** NEMO (Madec et al., 2008), SYMPHONIE (Marsaleix et al., 2008, 2009, 2012) **or** MARS3D (Lazure and Dumas 2008) ocean models are coupled to SURFEX" .*

**RC:** The next sentence is a bit too negative for my taste and I would skip it.

*AA: Done*

**RC:** If someone wants to add an exotic new variable to exchange, then it is obvious that the flow can change. But it is not necessary to discuss this here already.

*AA: The sentence has been removed and added in the conclusion ln 561.*

**RC:** Page 11, line 323, after time step, please add "is".

*AA: Done*

**RC:**Line 325 please change: domain size, integration duration and SURFEX version used.

*AA: Done*

**RC:** Page 12, line 383: lucia is only available since OASIS3-MCT3.

*AA: Done*

**RC:** Page 18, line 533 please change: the applications described here show that the computational cost of the OASIS3-MCT interface are negligible compared to the costs of the individual model components.

*AA: Done*

**RC:** Line 546: "fulfilled" is wrong, "edited" could be used

*AA: Done*

**RC:** Line 548: Ocean models are usually run either forced by fluxes or forced by near surface atmospheric fields. In the last case, bulk formulae are embedded in the ocean code to calculate turbulent fluxes. The comparison with coupled run is not straightforward in this case as bulk formulae used in the ocean code are not the same as those used in the coupled system.

I would suggest adding two paragraphs of the first version of the paper from the conclusions. Page 18, lines 546ff: it is often hard to clearly isolate… [continued on page 19] both in the ocean and atmospheric systems.

- Discussion around the benefits of a coupled system

- Improving the mixed layer in ocean and atmosphere

*AA: Done*

[revised manuscript text omitted]